



# The role of fuel and environmental conditions on the amount and composition of primary, fresh, and aged aerosol emissions originating from diesel- and gasoline-operated auxiliary heaters of passenger cars

Henri Oikarinen[1], Anni Hartikainen[2], Pauli Simonen[3], Miska Olin[4], Ukko-Ville Mäkinen[3], Petteri Marjanen[3], Laura Salo[3], Ville Silvonen[3], Sampsa Martikainen[3], Jussi Hoivala[3], Mika Ihalainen[2], Pasi Miettinen[1], Pasi Yli-Pirilä[2], Olli Sippula[2], Santtu Mikkonen[1,2], Panu Karjalainen[3,5]

[1]Department of Technical Physics, University of Eastern Finland, Kuopio, 70210, Finland
[2]Department of Environmental and Biological Sciences, University of Eastern Finland, Kuopio, 70210, Finland
[3]Aerosol Physics Laboratory, Physics Unit, Tampere University, Tampere, 33720, Finland
[4]Department of Atmospheric Sciences, Texas A&M University, College Station, TX 77843, United States
[5]Tampere Institute for Advanced Study, Tampere University, Tampere, 33720, Finland

*Correspondence to*: Henri Oikarinen (henri.oikarinen@uef.fi)

**Abstract.** Fuel-operated auxiliary heaters (AH) are potentially significant additional sources of particle and gas phase pollution from vehicles, but information on their emissions is scarce. Especially understanding of AH exhaust−originated secondary aerosol formation is lacking. In this study, we measured the gas and particle emissions, including secondary emissions, of diesel- and gasoline-operated AHs used in passenger cars. Investigation revealed the importance of peak emissions during start and shutdown events of the heaters and differences between emissions of gasoline and diesel fuelled

AHs. Namely, gasoline-operated AHs produced particles also under steady-state operating conditions, while their diesel counterparts did not. Furthermore, ambient air temperature was observed to impact the emission profiles, with, for example, higher $NO_x$ and particle mass emissions but lower particle number emissions observed in outdoor (−19 to −7 °C) measurements compared to laboratory measurements (+25 °C). However, further quantification is necessary to fully quantify the temperature-related effects to AH emissions. Our findings highlight the importance of characterizing also the

atmospherically aged aerosols, specifically secondary organic aerosol formation, which was here simulated both by an environmental chamber and by an oxidation flow reactor. The particle mass in photochemically aged aerosols surpassed the fresh exhaust particulate mass emissions by 1 to 3 orders of magnitude, with the increase depending mainly on fuel, combustion conditions, and aging methods. Further research into formation pathways of secondary aerosols from precursors is still needed as well as quantification of vehicle fleet level AH emissions to enable estimation of atmospheric and air

quality effects of AH usage.



## 1 Introduction

In recent decades, the automotive industry has witnessed a shift towards the integration of various auxiliary heating systems aimed at enhancing the comfort and functionality of vehicles in cold climates. One such system is a fuel-operated auxiliary heater (AH), a technology designed to provide supplementary heating to the vehicle cabin and engine. These systems, exemplified by the widely used heaters manufactured by Webasto, Eberspächer, and others, offer the convenience of pre-heating the vehicle interior while reducing the strain on the engine during initial cold ignition. This improves passenger comfort and claims reduced wear on critical engine components. The popularity of AHs has somewhat grown in response to the need for cabin and engine heating due to the increased efficiency of combustion engines and powertrain hybridization. Cold weather conditions not only create discomfort for occupants but also pose challenges to the usability, efficiency, and emissions of internal combustion engines, but this needs to be balanced against calculated environmental impact of AH usage. This has brought AHs into the spotlight of automotive engineering.

As emission regulations become progressively more stringent worldwide, there is a growing concern regarding the potential impact of AHs on air quality and climate. AHs introduce an additional source of transport sector pollutants into the atmosphere. The emissions from AHs, including particulate matter (PM), black carbon (BC), nitrogen oxides ($NO_x$), and carbon monoxide (CO), are of particular interest due to their harmful impacts for air quality and human health. The current AH emission regulation in use in the EU and some other countries limits only CO, $NO_x$, and hydrocarbon (HC) emissions with maximum concentrations under stable flame conditions being 1000, 200, and 100 ppm respectively (Regulation 122/2010). The literature covering the aerosol emissions from fuel-operated AHs is limited. Besides our recent studies (Karjalainen et al., 2021; Oikarinen et al., 2022), only few peer-reviewed studies on AH emissions (Nagy et al., 2024; Pettinen et al., 2023) exist. AHs have been observed to, e.g., produce over 3 orders of magnitude higher non-volatile particle number (PN) concentrations than diesel or gasoline engines on idle (Karjalainen et al., 2021). The environmental impact of AHs may be further impacted by atmospheric processes.

In addition to primary pollutants, such as PM mass, PN, and BC, gasoline and diesel combustion engines also generate precursors capable of forming secondary aerosols in the atmosphere. Interestingly, the quantity of secondary aerosol emissions can vastly surpass that of primary or fresh aerosol emissions (Hartikainen et al., 2023; Karjalainen et al., 2016; Kostenidou et al., 2024), especially in case of PM mass. Fuel composition influences precursor emissions (Karjalainen et al., 2019; Timonen et al., 2017), which subsequently influence the formation of secondary organic aerosol (SOA). However, SOA emissions have not yet been reported for AH exhaust.

Aims of this study are to determine emission factors of primary and secondary aerosol pollutants and gas phase pollutants from AHs. Emissions of gas phase pollutants are also compared against both regulation limits and mean fleet emission factors of EU road transport emissions. We strive to understand how both ambient temperature and atmospheric



photochemical aging influences AH exhaust aerosols. Further we aim to validate AH emissions aged with oxidation flow

reactor against those emissions aged with more realistic atmospheric oxidation chamber. Chamber experiments also provide

insight onto the time scales of the aging process.

## 2 Material and methods

Experiments conducted for this research consist of outdoor and laboratory measurement campaigns. The outdoor study that

was carried out under real winter conditions in Finland, employing partial exhaust sampling and an oxidation flow reactor

(OFR) system to capture the evolving nature of emissions in real-world scenarios. The laboratory investigation, on the other

hand, employed a large atmospheric oxidation chamber of University Eastern Finland that allows controlled and replicable

batch-type aging measurements. Both measurement campaigns focused on a broad range of emissions parameters measured

in real time including various particle and gaseous emissions, particle number and mass, black carbon concentration, and

chemical composition analysis.

### 2.1 Laboratory measurement campaign (ILMARI)


#### 2.1.1 Sampling and experimental Setup

The indoor laboratory measurement campaign was conducted within the ILMARI laboratory of University of Eastern

Finland using a Euro6-compliant gasoline vehicle (Skoda Octavia) and a Euro5b diesel vehicle (Seat Alhambra) that were

both equipped with 5 kW Termo Top Evo fuel-operated auxiliary heaters manufactured by Webasto. The gasoline vehicle

AH was operated on commercially available 95E10 fuel (EN228, 95 octane, max 10% ethanol). For the diesel AH

experiments, two different fuel types were used: a renewable diesel produced by hydrotreatment of waste oils and fats (Neste

MY Diesel, EN15940) and more conventional fossil-based diesel (Futura, EN590). Tests were conducted indoors at approx.

25 °C on different days, giving the AHs sufficient time for cooling back to room temperature overnight after the previous

experiment was finished.


The test procedure consisted of an approximately 20-min period of AH operation, during which the AH exhaust was sampled

with on-line instrumentation to provide a time series of emissions. A partial flow of the exhaust was introduced to an

environmental chamber (Sect. 2.1.3) simultaneously. When the AH running period was finished, the online instrumentation

sampling point was switched from the exhaust pipe to the environmental chamber with a 3-way valve. The complete setup of

the laboratory experiments is illustrated in Fig. 1a.

The exhaust was transferred with a heated line (250 °C), from which the sample was divided into the on-line instrumentation

and to the environmental chamber. The sample for on-line instrumentation was diluted and cooled down with a combination

of a porous tube diluter (PTD) and an ejector diluter (ED). The PTD dilution air temperature was controlled at 30 °C and the



dilution flow was adjusted to obtain a dilution ratio (DR) of 12. The PTD was followed by a residence time tube with a residence time of 2.5 s, after which the sample was further diluted with an ejector diluter. These parameters of the PTD have been shown to mimic the atmospheric dilution process well in a real-world driving situation (Keskinen and Rönkkö, 2010).

A partial flow of the exhaust sample downstream of the PTD and the ejector was led through an OFR to provide time series
of the secondary aerosol formation potential of the transient exhaust. The OFR operation is described in Sect. 2.1.3.

**2.1.2 Photochemical processing in the chamber**

The laboratory campaign utilized a 29 $m^3$ Teflon fluorinated ethylene propylene (FEP) chamber (Leskinen et al., 2015) for tracing the atmospheric aging process of AH emissions in well characterized conditions. The chamber was surrounded by UV blacklights centered at a wavelength of 340 nm. The chamber has a moving top frame, intended to maintain slight
constant overpressure to prevent sample dilution and contamination with air outside the chamber during an experiment. This controlled setting allowed for the precise simulation and measurement of AH emissions, providing invaluable insights into emissions' behavior, composition, and transformations during atmospheric processing.

Exhaust emission was sampled to the chamber throughout the AH operation and diluted by a factor of 8 prior to chamber
input by a two-step dilution system consisting of a PTD and an ED. As the chamber had been pre-filled with clean air, the total DRs in the chamber were in the range of 640 to 720. After a stabilization and primary measurement period (30 to 35 min in total), $O_3$ was injected to the chamber to oxidize NO to $NO_2$ before initiation of the photochemical aging. $O_3$ concentrations in the chamber remained below 25 ppb after the $NO_2$ conversion. After the NO oxidation the UV lights were turned on and the chamber was operated in a batch-mode for 180 to 280 min. In the diesel AH experiments and in one
gasoline AH experiment, 0.5 mL of $H_2O_2$ in 30 % solution in water was evaporated into clean air and injected to the chamber as precursor for hydroxyl radicals (OH·) to guarantee an adequate degree of OH· exposure. Temperature in the chamber was 21 ± 1.2 °C and relative humidity 59 ± 5.3 % during the AH aging experiments.

The photochemical age in the chamber was determined based on the decay in the reactive gases (Barmet et al., 2012). For the
gasoline AH experiments, the photochemical age in the chamber extended to up to 0.2 atmospheric equivalent days at the average ambient OH concentration of 1.5 x $10^6$ $cm^{-3}$ (Fig. S1). A notably higher exposure (up to 2 atmospheric equivalent days) was reached during the diesel vehicle experiments due to the notably lower amounts of reactive components that consume the OH in the chamber.

Particle wall losses in the chamber were estimated using the polydisperse particle loss rate previously measured for the chamber (Leskinen et al., 2015). Fates of the low-volatility organic compounds (LVOCs) in the chamber were assessed as in the studies by Palm et al., 2016 and Hartikainen et al. 2023. For the diesel experiment with lowest condensation rate to



particles, the LVOC loss rates to the chamber walls were estimated to be in the same range as their condensation rates to particles. For the gasoline experiments, in contrast, the role of the wall losses was minor due to the higher level of particulate

condensation sink present (Fig. S2). Higher particulate condensation sink of gasoline experiments is due to higher initial PM concentrations in the chamber. Consequently, during gasoline experiments the LVOCs are expected to have condensed on the particle surfaces within minutes. See Supplementary Sect. S1 for detailed consideration of the chamber conditions.

### 2.1.3 Photochemical processing in the TSAR

The Tampere Secondary Aerosol Reactor (TSAR) (Simonen et al., 2017) facilitated the assessment of the secondary aerosol
potential in AH exhausts in three different scenarios: 1) directly from the AH tailpipe during the sample input period, providing time-resolved secondary aerosol potentials for AH operation; 2) from the stabilized exhaust emission from the chamber, before the photochemical aging, giving insights to the average potential in the exhaust; and 3) at the end of the chamber experiments (after 3–5 h), in order to obtain the unrealized secondary aerosol formation potential.

TSAR is an oxidative flow reactor (OFR) with UV lamps with an emission spectrum centered at 254 nm. It is optimized for monitoring the secondary aerosol potential in transient vehicle exhausts by having a near laminar flow and relatively short residence time (40 s). The photochemical ages of TSAR-aged exhausts were assessed to be in the range of 4.5–6.8 atmospheric equivalent days, based on the decrease in CO concentration caused by TSAR when its UV-lights were turned on. The TSAR was also used in the outdoor campaign (Sect. 2.2) with same settings.

### 2.1.4 Instrumentation and measurements

The measurement instrumentation for the laboratory campaign aimed to capture various aspects of the emitted aerosols' characteristics, including both the particulate and gaseous phases. Gaseous pollutants, including $CO_2$, CO, and a range of volatile organic compounds (Supp. Table 1) in the directly emitted exhaust aerosol were measured by a Fourier Transform Infrared Spectrometer (FTIR, Gasmet Technologies, Inc.). The concentrations of the sample total organic gaseous
compounds (OGC) were calculated from the measured organic subspecies of methane, alkane, unsaturated HC, aromatic HC, oxygenated HC, and oxygenated aromatic HC groups. The OGC concentration (in ppmC) was calculated by weighing the concentrations of individual species by their respective number of carbon atoms per molecule. The $NO_x$ concentrations were measured with a chemiluminescence-based analyser (Model T200; Teledyne Instruments, Inc.). A VOCUS-PTR (Aerodyne, specs) was used for the validation of the photochemical aging of chamber experiments.


Particle number and size distributions were measured by two scanning mobility particle sizers (SMPS) systems, covering size ranges of 8.2 to 346 nm ("long-SMPS", TSI DMA 3071, TSI CPC 3775) and 3.5 to 64 nm ("nano-SMPS", TSI DMA 3085, TSI CPC 3776). The particle size distributions (PSDs) measured by the two instruments were combined by calculating weighted mean for the size bins covered by both instruments (8.2 to 64 nm). Weighting coefficients were obtained from a





cubic fit from 1 to 0 with results from nano-SMPS emphasised for small end of overlapping PSD, with initial weight of 1 for

8.2 nm size bin and results from long-SMPS emphasised for large end of PSD. Geometric mean diameters (GMDs) were

calculated from the combined PSDs. It should be noted that the SMPSs were operated on a time resolution of 3 minutes per

scan over the whole PSD. The rapid changes that occurred within roughly 1−2 minutes during initial ignition and shutdown

phases of the AH heating cycle were thus not fully captured by the SMPS. Further, since the SMPSs scanned the particle size

bins from smallest to largest, there is a misalignment in the true underlying PSDs and the one measured with SMPS, as there

is over minute of time difference between first and last size bin measured during scan.

A condensation particle counter (CPC) battery, consisting of four CPCs with different cut-off particle diameters, was

employed to determine particle number concentrations with information on particle size distributions. Number of particles

above the mobility diameters of 23 nm, 10 nm, and 2.5 nm, were measured by Airmodus A23, Airmodus A20 and TSI 3756,

respectively, whereas the total number of particles larger than 1.3 nm was monitored by a Particle Size Magnifier (PSM,

Airmodus) coupled with the Airmodus A20 CPC. Same CPC battery was also used in outdoor measurement campaign.

However, it should first be noted that differences in sampling systems used affect the effective cut-off diameters of CPCs,

with effective cut-off diameter here defined as the particle diameter of which 50 % are detected after sampling line losses are

accounted for. Effective cut-off diameters for CPCs are presented for both campaigns in Fig. S7 and based on it we note that

for CPCs with cut-off diameter of 1.3, 10 and 22 nm effective cut-off diameters match well for both campaigns and are thus

comparable. Whereas for the CPC with cut-off diameter of 2.2 nm difference in effective cut-offs is significant with it being

3.7 nm and 6.3 nm in outdoor measurements and laboratory measurements respectively, due to which we present comparison

of PN values from that CPC in table 2 but exclude it from further EF analysis.


An aethalometer (AE33, Aerosol Magee Scientific) was used to measure the equivalent black carbon (eBC) mass

concentrations and the wavelength dependency of light absorbance. eBC was quantified from the optical attenuation of light

at 880 nm assuming a mass absorbance cross-section of 7.77 m2/g and the instrument standard multiple-scattering correction

factor of 1.39. Absorption Ångström exponents (AAE) were calculated from the absorption of light at the wavelengths of

470 nm and 950 nm.

Two electrical low pressure impactors (ELPIs, Dekati) were utilized to determine the mass concentrations of fresh and aged

exhaust. An ELPI+ (Järvinen et al., 2014) measured the concentration upstream of the TSAR, which was sampling either

from the Teflon chamber or diluted exhaust. Further, an ELPI (Keskinen et al., 1992) measured the concentration

downstream of the TSAR. The ELPI was equipped with a filter stage (Marjamäki et al., 2002) and an additional impactor

stage (Yli-Ojanperä et al., 2010) so that its size bins matched the ELPI+ as closely as possible. The small particles generated

in the TSAR caused difficulties when inverting the measured current distribution into mass size-distribution. A fraction of



the small particles deposit onto the impactor stages representing larger particles due to diffusion (Virtanen et al., 2001). Thus, the inversion required special treatment which is described in the Supplementary Sect. S2.

## 2.2 Outdoor measurement campaign

The outdoor measurement campaign was built upon the experiments outlined in the study by Oikarinen et al., 2022. The objective was to capture AH emissions under real-world conditions, providing crucial data for assessing emissions in natural environments. The campaign incorporated the TSAR for the photochemical aging of the exhaust.

### 2.2.1 Test vehicles

Information about the vehicles, their AHs and the measurement setup is presented in the study by Oikarinen et al., 2022. The test matrix included six vehicles in total, three of which were operated with gasoline and three with diesel. However, here we focus on the same two vehicles, Skoda Octavia and SEAT Alhambra, that were also used in the laboratory campaign. The used gasoline fuels were of the 95E10 grade, and the diesel fuels met the EN 590 standard. The AHs used the same fuel as the vehicles they were installed in.

### 2.2.2 Environmental conditions, sampling strategy and instrumentation

The tests were conducted within four consecutive days under Finnish winter conditions (−19 to −7 °C), and each AH was measured four times. The measurement protocol included an overnight cooling period of at least 18 h, after which the vehicle was driven from the nearby parking slot to the sampling location (both outdoors, open air). This transfer was kept as brief as possible (less than 20 s), to prevent excessive engine heating and automatic starting of the AH typically occurring after roughly 1 min of driving.

After attaching the sampling line to the exhaust pipe of the AH, the heater was turned on and left to run for about 30 min. Both the total run time and the general operation pattern of the AHs varied between the vehicles. Gasoline AH ran steadily for the predetermined heating time until manual shutdown, whereas diesel AH sometimes switched to half power for few minutes during end of preheating due to heater reaching its nominal target temperature of 67 °C. Parameters describing the operation of the heater were recorded using on-board diagnostics.

The outdoor campaign employed a partial sampling strategy to directly capture emissions from vehicles equipped with AHs. The collected emissions were analysed for PM and PN concentration, gaseous pollutants, and black carbon content. The inclusion of a TSAR, allowed for the investigation of emissions' secondary aerosol formation potential and aging effects on chemical composition of particles. The photochemical age in TSAR was in the range of 4.6−5.9 days, based on a decrease in CO concentrations inside TSAR when the UV-lights were turned on.





### 2.2.3 Chemical composition analysis

The chemical composition of the emissions was measured using the Soot Particle Aerosol Mass Spectrometer (SP-AMS; Aerodyne Inc.; Onasch et al., 2012) with which half of the measurements were measured with the TSAR (aged exhaust) and half without TSAR (fresh). The SP-AMS analysis provided critical insights into the chemical composition of emitted particles, enhancing the understanding of the potential environmental impacts of AH emissions. The SP-AMS is based on the HR-ToF-AMS (DeCarlo et al., 2006) with a SP-module for detecting refractory species including refractory black carbon (rBC) and some metals. The transmission efficiency for particles in the size range of 50 to 1000 nm is reasonable, while smaller and larger particles can be lost in the aerodynamic lens (Liu et al., 2007). A rotating chopper can be operated in open, closed or PToF position, allowing particles to pass through it, not allowing them to pass or allowing them to pass periodically, respectively. This is used for determination of the background signal and for size resolved composition analysis of the particles. The sample is vaporized using a tungsten vaporizer (600 °C) and a Nd:YAG laser vaporizer (1064 nm) and ionized with 70 eV electron ionization. (DeCarlo et al., 2006)

The data-analysis was performed in PIKA (v.1.25) and SQUIRREL (v.1.65) using Igor Pro 9 (Wavemetrics Inc.). The ionization efficiency (IE) and relative ionization efficiency for black carbon (RIEBC) were measured using methods described by (Onasch et al., 2012) and have been applied to the data. Composition dependent collection efficiencies (CDCE) have also been calculated and applied to the data (Middlebrook et al., 2012). To simplify the analysis of the hundreds of observed compounds, the results are divided into families of ammonium, chloride, nitrate, organic matter, refractory BC, and sulfates.

A flame ionization detector (FID; Mocon Baseline 9000) was used for quantification of total gaseous hydrocarbon (THC) content. It should be noted that THC measured with FID in the outdoor campaign and OGC from FTIR in the laboratory campaign are both approximations of true gaseous organic content of exhausts due to FID's low sensitivity to oxygenated HCs and formaldehyde and due to FTIR not exhaustively detecting all gaseous organic species. Nevertheless, a review of studies comparing FTIR-measured HC content to FID for vehicle exhaust found out that differences typically fall within 15% range, but underestimation by 50 % and overestimation up to 34 % have also been reported (Giechaskiel et al., 2021).

### 2.3 Emission factor calculations

Emission factors (EF) of AHs were calculated for 1) a 30 min of preheating (EF$_{30min}$), 2) per 1 min of stable operation (EF$_{min}$), 3) EF per kg of fuel consumed during preheating (EF$_{fuel}$), and 4) fraction of total emission produced during ignition and shutdown spikes for both outdoor and laboratory measurement campaigns. EFs were calculated using integral method outlined in (Oikarinen et al., 2022). Total emission $E$ is calculated with Eq. (1):



$$E \;=\; \frac{\int_0^t [x] dt}{\int_0^t [CO_2] dt} Kt\rho \frac{m_{C,fuel}}{m_{tot,fuel}} \frac{M_{CO_2}}{M_C},$$  (1)

where $[x]$ (kg or particle per m3) is measured concentration of substance, $t$ (h) is duration of heating cycle, $K$ (l h$^{-1}$) is fuel consumption rate of AH, $\rho$ is the density of the fuel (0.740 kg l$^{-1}$ for gasoline and 0.805 kg l$^{-1}$ for diesel), $m_{C,fuel} m^{-1}_{tot}$, fuel is carbon mass fraction of fuel and $M_{CO_2} MC^{-1}$ is ratio of mass increase when carbon in fuel is transformed into $CO_2$ and released in combustion. From the total emission $E$ emission factor relative to fuel consumption were calculated by simple

division of $E$ with mass of consumed fuel. EFs relative to heating time of either 30 or 1 min are simply total emission $E$ for respective duration of AH heating. EF$_{min}$ was calculated by excluding ignition and shutdown phases from the time series of the measurement and calculating mean EF for 1 min of emissions.

Fraction of total emissions produced during ignition and shutdown were further calculated by comparing $E$ over only

ignition and shutdown phases. Ignition and shutdown phases are defined here as time it takes $CO_2$ to change from background level to stable operating levels for ignition and reverse for shutdown phase. For AHs in used in these measurements ignition and shutdown phases correspond approximately to first 2.5 minutes and last 2 minutes of AH operation, respectively.

EFs from these two campaigns are compared to detect how operating temperature affects the emission profile of the AH during operation. For the laboratory measurements EFs of exhaust sample aged with ILMARI chamber are also calculated to account for the effects of secondary aerosol formation.

Fresh particle emissions were compared to aged emissions in both outdoor and laboratory measurements. Fresh emission

factors were calculated either directly during from emissions AH heating cycle or from concentration in the chamber at the beginning of 4 h aging process. Aged emission factors were calculated by multiplying fresh EF by ratio of (TSAR and/or Chamber) aged emission concentrations to fresh concentrations measured from the environmental chamber, example of which is provided in Fig. S10.

It should be noted that emission factors were calculated based on the assumption that fuel consumption and exhaust air flow are constant, as neither was measured in real time. These assumptions systematically overestimate the contribution of both ignition and shutdown emission spikes to EFs of a single preheating cycle. During ignition, it takes a while for the air flow out of the AH to stabilize. During the shutdown phase, after fuel consumption has ended, the airflow out of the AH decreases while concentrations of exhaust emissions increase as can be seen from Figs. 2−6. Thus, when we convert concentrations to

EFs we overestimate these emission spikes due to an overestimation of the exhaust flow.



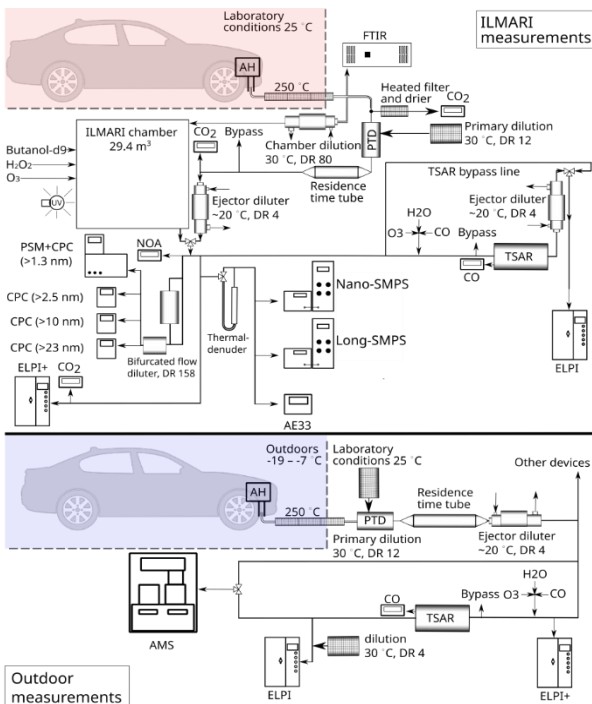

**Figure 1: Measurement setup in the ILMARI research environment and the outdoor research environment.**

## 3 Results and discussion

### 3.1 Gaseous emissions

#### 3.1.1 Carbon dioxide emissions and implications on emission factors

Temporal variation in $CO_2$ concentrations presented in Fig. 2 showed a steep increase during first roughly 2.5 minutes from ignition, constant concentrations during the stable operating period, and a steep decrease during the shutdown phase lasting roughly 2 minutes. This behavior was, in general, similar in both laboratory and outdoor measurements. In one of the outdoor experiments, the AH switched to half power near the end of operation after sufficiently high coolant temperature had been reached, causing a clear decrease of $CO_2$ concentrations Fig. 2d.

The mean exhaust $CO_2$ concentrations were different in laboratory and outdoor conditions, with 21 % and 37 % higher concentrations under laboratory conditions for gasoline AHs and diesel AHs, respectively. This indicates variation in the air−fuel ratio (AFR) of combustion, caused likely by temperature dependency of the amount of combustion air injected into combustion chamber of AH. Since assuming a constant volumetric flow of air into the combustion chamber, the mass flow of air was greater by a factor of 1.12−1.17 in sub-zero outdoor conditions (when 25 °C indoor air is compared to –7 or –19 °C outdoor air), leading to a higher AFR which explains most of the difference in $CO_2$ concentrations in exhaust gas in cold





outdoor operating conditions. Another explanation for the varying $CO_2$ concentrations would be variation in fuel
consumption rate, which was not directly measured for these experiments. However, the fuel consumption of AHs was
estimated to be constant irrespective of the temperature outside the vehicle in a separate set of experiments (Ala-Hakuni et
al., in prep.). For laboratory measurements, Changes in chemical composition in renewable diesel compared to regular diesel
might also have caused slight decrease in $CO_2$ concentrations, due to renewable diesel containing larger fraction of aliphatic
hydrocarbons, which under ideal stoichiometric conditions is expected to produce lower $CO_2$ concentration to exhaust.


The differences in $CO_2$ concentrations between the campaigns also directly impacted the ratios of EFs to exhaust
concentrations. As change in AFR affects both $[CO_2]$ and $[x]$ and thus calculated EFs are not as affected by the changes in
AFR as concentrations. Due to this even if almost equal concentration for $[x]$ is observed between measurement campaigns it
does not result in equal EFs.

**3.1.2 Inorganic gaseous emissions**

Timeseries of inorganic gaseous emissions are presented in Fig. 2, for $CO_2$ and $NO_x$ emissions from both measurement
campaigns, and additionally for carbon monoxide (CO), sulfur dioxide ($SO_2$), Ammonia ($NH_3$), hydrogen chloride (HCl),
and hydrogen cyanide (HCN) emissions for the laboratory campaign where FTIR data was available. For these gaseous
emissions, mean concentrations, EFs, and spike fractions of total emissions are presented in Table 1, with spike fraction
defined as fraction of total emission produced during ignition and shutdown period. Carbonyl sulfide (COS) and hydrogen
fluoride (HF) emissions were also measured during the laboratory campaign and found to be below detectable levels for all
AH tests and are thus omitted from Table 1 and Fig. 2.

For gasoline AHs (Fig. 2a-b), CO emissions exceeded the regulation limit of 0.1 % for most of the preheating cycle. For the
diesel AHs, CO emissions remained below the regulation limit after a brief exceedance during the ignition spike (Fig. 2c-d).
$EF_{fuel}$s for CO showed clear differences between fuel types with gasoline and diesel AHs producing on average 21 and 4.2 g
CO $kg_{fuel}^{-1}$, respectively (Table 1). If CO emissions of AHs are compared against driving emissions of cars, with estimated
mean EU fleet $EF_{fuel}$s for road transport being 48.36 (gasoline), and 2.41 (diesel) g $kg_{fuel}^{-1}$, then gasoline AH produces more
CO than road transport and diesel AH produces less CO. If CO emissions of AHs are instead measured against newer euro 5
compatible vehicles with estimated mean $EF_{fuel}$s of 8.60 (gasoline), and 0.52 (diesel) g $kg_{fuel}^{-1}$ then AH emissions are
significantly higher than emissions of newer vehicles for both gasoline and diesel (Ntziachristos, 2024).

$SO_2$ emissions of gasoline AH were almost exclusively released at ignition and shutdown, with only short bursts of $SO_2$
detected during the stable operating period (Fig. 2a-c). The diesel AH operated with regular diesel produced $SO_2$ similarly to
the gasoline AH (Fig. 2d). Instead, when using renewable diesel, the concentration of $SO_2$ was detectable during the whole





preheating cycle. This is also reflected in the $SO_2$ $EF_{fuel}$s where diesel, on average, produced an order of magnitude more of $SO_2$ emissions, with 5.2 and 58 mg of $SO_2$ per kilogram of fuel for gasoline and diesel, respectively.

$NH_3$ emissions of gasoline and diesel AHs lacked detectable ignition spikes with most of emissions produced at steady rate during whole preheating cycle. Further notable shutdown spikes of $NH_3$ were detected for gasoline AH. Mean $NH_3$ $EF_{fuel}$s were 0.73 and 0.51 g $kg_{fuel}^{-1}$ for gasoline and diesel, respectively. Compared against driving emissions, these $EF_{fuel}$s are significantly lower than even the lower estimate of Euro5-compatible EU vehicle fleet mean $EF_{fuel}$s: 20, and 10 g $kg_{fuel}^{-1}$ for gasoline and diesel, respectively (Ntziachristos, 2024).

The $NO_x$ emissions were stable for almost all measurements and stayed below the regulation limit of 200 ppm. A slight increasing trend in $NO_x$ concentrations during the preheating cycle was observed. The $NO_x$ emissions lacked the distinctive ignition and shutdown spikes observed for other measured pollutants, except for the first gasoline AH measurement where the $NO_2$ concentration momentarily exceeded 5000 ppm at shutdown. This is, however, acceptable by the regulation since the spike did not occur under stable flame conditions but coincided with decreasing $CO_2$ concentration typical for shutdown
period. The $NO_x$ EFs for both gasoline and diesel AHs were smaller by roughly one-third under laboratory conditions compared to outdoor conditions (Table 1).

**Table 1. Mean concentrations, emission factors per kg of consumed fuel, for the whole 30-minute cycle, for 1 minute of stable operation, and the fraction of emission emitted during the spike periods (ignition and shutdown) for the**
**gaseous emission, with standard deviations of respective values inside brackets. Measurements for which measured values were below device detection limit are marked with BD, whereas – indicates data not being available.**

| Quantity (device) | Unit | Gasoline AH | | | Diesel AH | | |
|---|---|---|---|---|---|---|---|
| | | Outdoor | Laboratory | Ratio between laboratory and outdoor | Outdoor | Laboratory | Ratio between laboratory and outdoor |
| CO (FTIR) | ppm | - | 1100 (67) | - | - | 160 (34) | - |
| | g kg fuel$^{-1}$ | - | 21 (1,1) | - | - | 4,2 (0,76) | - |
| | g 30min$^{-1}$ | - | 5,6 (0,29) | - | - | 1 (0,19) | - |
| | g min$^{-1}$ | - | 0,19 (0,011) | - | - | 0,028 (0,0053) | - |
| | Spike % | - | 9,4 (0,61) | - | - | 23 (0,27) | - |
| SO$_2$ (FTIR) | ppm | - | 0,12 (0,099) | - | - | 0,97 (0,69) | - |
| | mg kg fuel$^{-1}$ | - | 5,2 (4,3) | - | - | 58 (42) | - |
| | mg 30min$^{-1}$ | - | 1,4 (1,1) | - | - | 14 (11) | - |
| | mg min$^{-1}$ | - | BD | - | - | 0,39 (0,32) | - |
| | Spike % | - | 100 (0) | - | - | 32 (14) | - |




| | | | | | | | |
|---|---|---|---|---|---|---|---|
| **NH₃ (FTIR)** | ppm | - | 0,065 (0,054) | - | - | 0,032 (0,017) | - |
| | mg kg fuel⁻¹ | - | 0,73 (0,62) | - | - | 0,51 (0,29) | - |
| | mg 30min⁻¹ | - | 0,19 (0,16) | - | - | 0,13 (0,072) | - |
| | mg min⁻¹ | - | 0,004 (0,0023) | - | - | 0,003 (0,0019) | - |
| | Spike % | - | 32 (31) | - | - | 33 (0,93) | - |
| **HCl (FTIR)** | ppm | - | 0,27 (0,054) | - | - | 0,26 (0,037) | - |
| | mg kg fuel⁻¹ | - | 6,6 (1,3) | - | - | 8,9 (1,5) | - |
| | mg 30min⁻¹ | - | 1,7 (0,33) | - | - | 2,2 (0,37) | - |
| | mg min⁻¹ | - | 0,052 (0,005) | - | - | 0,074 (0,012) | - |
| | Spike % | - | 17 (8,4) | - | - | 7 (0,87) | - |
| **HCN (FTIR)** | ppm | - | 1 (0,2) | - | - | 0,84 (0,22) | - |
| | mg kg fuel⁻¹ | - | 18 (3,8) | - | - | 21 (5) | - |
| | mg 30min⁻¹ | - | 4,8 (1) | - | - | 5,2 (1,2) | - |
| | mg min⁻¹ | - | 0,16 (0,041) | - | - | 0,18 (0,053) | - |
| | Spike % | - | 11 (3,2) | - | - | 8,1 (2,2) | - |
| **NOₓ (NOA)** | ppm | 100 (5,9) | 80 (0,58) | 0,79 (0,047) | 68 (4,5) | 65 (4,4) | 0,96 (0,091) |
| | mg kg fuel⁻¹ | 4100 (110) | 2500 (0,74) | 0,61 (0,016) | 3900 (47) | 2800 (110) | 0,71 (0,03) |
| | mg 30min⁻¹ | 1100 (41) | 650 (0,19) | 0,61 (0,023) | 940 (30) | 690 (28) | 0,74 (0,038) |
| | mg min⁻¹ | 36 (1,8) | 22 (0,12) | 0,61 (0,03) | 31 (0,61) | 23 (1) | 0,75 (0,035) |
| | Spike % | 13 (1,3) | 7,5 (0,16) | 0,57 (0,056) | 14 (1,9) | 7 (0,13) | 0,51 (0,072) |
| **THC/OGC (FID/FTIR)** | ppm | 70 (11) | 140 (110) | 2 (1,6) | 76 (2,5) | 77 (23) | 1 (0,31) |
| | mg kg fuel⁻¹ | 750 (140) | 1100 (870) | 1,5 (1,2) | 1100 (40) | 850 (240) | 0,77 (0,21) |
| | mg 30min⁻¹ | 200 (32) | 300 (230) | 1,5 (1,2) | 270 (9,5) | 210 (59) | 0,8 (0,22) |
| | mg min⁻¹ | 3,6 (0,19) | 2,4 (0,0057) | 0,65 (0,034) | 6,3 (0,62) | 3,5 (0,37) | 0,55 (0,08) |
| | Spike % | 35 (5,8) | 69 (24) | 2 (0,75) | 31 (8,7) | 52 (9,9) | 1,7 (0,57) |

### 3.1.3 Organic gaseous emissions

The concentrations of THC (outdoor) or OGC (laboratory) remained well within the regulation limit of 100 ppm during stable period of AH operation, although this limit was exceeded during ignition and especially shutdown phases for nearly all test cases (Fig. 2). On average, the measured THC and OGC concentrations were similar for diesel AHs in outdoor and indoor conditions, with the main differences being temporal distribution of emissions during the preheating cycle. Compared to the outdoor measurements, the laboratory experiments produced higher concentrations of organic gases during ignition and shutdown, but stable operating period concentrations were lower. The gasoline AH produced twice the THC or OGC concentrations of the diesel AH on average, in both measurement campaigns. The temporal distribution was similar for all the cases, with shutdown spike in the gasoline test 1 in Fig. 3b in being a major outlier with a 1 order of magnitude higher shutdown spike concentration when compared to gasoline test 2. When outdoor emission factors for THC are compared





against OGC emission factors in laboratory as presented in Table 1 show a relative increase by factor of 1.5 for gasoline and decrease by factor of 0.77 for diesel. Based on these results and known uncertainties between the FID-measured THC and FTIR-measured OGC, it cannot clearly be concluded if the operating temperature significantly affect the organic gaseous

emissions of AHs if the most extreme estimates for FTIR-measured OGC and FID-measured THC differences are used, especially, when large variations in spike concentrations are also major source of uncertainty.

Chemical compositions of the OGCs from the indoor measurements are presented in Fig. S5. During the stable period, the unsaturated HCs were the most abundant component of the OGC emissions. The concentration of unsaturated hydrocarbons

increased little during ignition and shutdown, when compared to alkane, aromatic and oxygenated HCs, which consequently dominated the organic gaseous emissions of the gasoline during ignition and shutdown periods. For diesel AH, alkanes, aromatic hydrocarbons, and unsaturated hydrocarbons were had similar concentrations during stable period and during the shutdown spike oxygenated HCs overtake unsaturated ones.

Renewable diesel was observed to reduce the overall levels of OGC by 38% compared to regular diesel. The type of diesel fuel used seems to also affect composition during the ignition phase with aromatic HCs replacing alkanes as the second most common HCs when regular diesel is used when compared to renewable diesel.





Figure 2: Concentrations of inorganic gaseous substances measured by FTIR in the laboratory measurements from the raw exhaust gas and with FID for outdoor NO+NO₂ for gasoline experiments (a-b), renewable diesel (c) and regular diesel (d). Shutdown of the AH at end of the preheating cycle is indicated with the black vertical dash lines and for CO regulation limit of 1000 ppm is marked with red horizontal dash line.



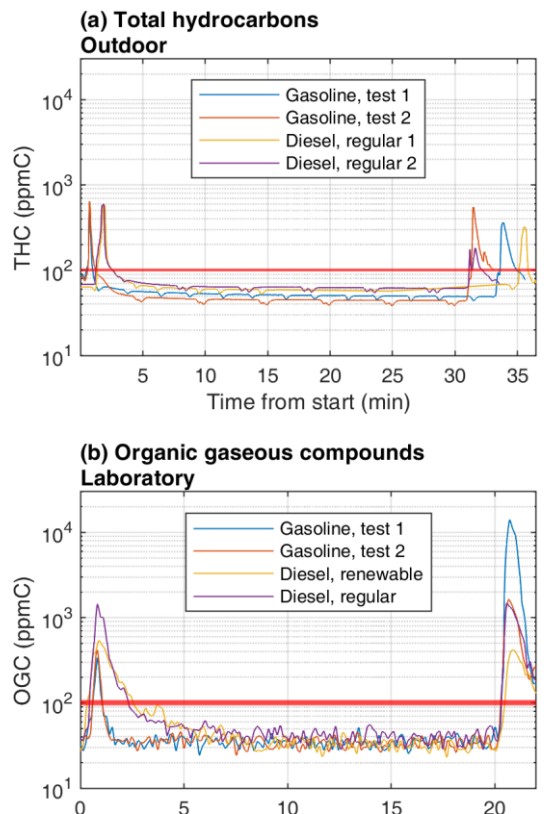

**Figure 3: Total gaseous hydrocarbons (THC) or organic gaseous compounds (OGC) concentrations in the AH exhausts directly measured by FID for the outdoor (a) and by FTIR for the laboratory measurements (b). The regulation limit of 100 ppmC for gaseous hydrocarbons is illustrated with the red horizontal lines.**

### 3.2 Transient Particle emissions

### 3.2.1 Particle number

PN emissions from gasoline AHs were considerably higher than diesel AH. During the laboratory measurements, gasoline AH produced average PN concentrations of $56 \times 10^6$ cm$^{-3}$ and $12 \times 10^6$ cm$^{-3}$ for >3.4 nm and >22 nm particles, respectively during the preheating, whereas diesel AH produced on average $1.5 \times 10^6$ cm$^{-3}$ and $0.27 \times 10^6$ cm$^{-3}$ for >3.4 nm and >22 nm particles, respectively. For diesel AH, the emissions of the preheating were concentrated on ignition and shutdown spikes (99 % of total PN emissions) with low stable period concentrations in the range of $10^2$ to $10^4$ cm$^{-3}$ (Fig. 4). Interestingly, there was also difference in the magnitude of ignition and shutdown spikes between diesel fuel types. For renewable diesel, the ignition spike is smaller with concentration of $0.14 \times 10^8$ cm$^{-3}$ than for regular diesel of $0.5 \times 10^8$ cm$^{-3}$. For the shutdown spike, the effect of fuel was reversed with renewable diesel producing larger shutdown spike of $1.3 \times 10^8$ cm$^{-3}$ when compared to $0.7 \times 10^8$ cm$^{-3}$ for regular diesel. Gasoline AH also produced PN spikes during ignition and shutdown, but it had





higher stable operating emissions with 70 % of the total preheating-originated PN emitted during the stable operating period and the ratio of the spike concentrations to stable ones being in the range of 2-3. Despite the type of the gasoline fuel used was the same in both gasoline tests, there are still major differences in the PN concentrations, with the stable operation concentrations being 2.0 times greater in test 2 when compared to test 1 and ignition and shutdown spikes were 2.3 times greater. The major difference in PN values of tests 1 and 2 are mostly explained by differences in sub-11 nm particles since if PN>11 nm values are compared they remain relatively similar between the tests. The ratios of the mean concentrations of test 2 to test 1 PN>11 nm are 0.86 and 0.94 for stable period concentrations and spike concentrations, respectively.

The AH PN emissions in outdoor PN conditions have already been published in (Oikarinen et al., 2022). In laboratory measurements similar features in the time series of PN was observed between the diesel and gasoline AHs as was observed in outdoor measurements with clear ignition and shutdowns spikes and low stable period emissions for diesel AHs. For the gasoline AH major differences in PN concentrations in outdoor compared to indoor measurements were detected with the ratio of laboratory to outdoor mean PN concentrations being 6.7, 4, and 4.3 for cut-off diameters of 3.4, 11, and 22 nm, respectively. Meanwhile, results for the diesel AH showed a shift in particle size distribution with an increase in sub-11 nm particle concentration, with corresponding reduction in larger particles. The ratios of laboratory to outdoor mean PN being 1.7, 0.64, and 0.95 for cut-off diameters of 3.4, 11, and 22 nm, respectively for diesel AH.

For the gasoline AH, PN emission factors were clearly higher in the laboratory compared to the outdoor conditions. Namely, the $EF_{fuel}$s were 5.1, 2.8, and 3.3 times higher for 3.4 nm, 11 nm, and 22 nm effective cut-off diameters, respectively. For the stable operating periods of the gasoline AHs, the difference between outdoor and laboratory conditions was even greater, with factor of 5.5, 3.3, and 3.5 increase in PN $EF_{fuel}$s for 3.4 nm, 11 nm, and 22 nm cut-offs, respectively. Consequently, the relative contribution of ignition and shutdown spikes to the total preheating cycle emissions was 60 % lower in laboratory. Compared to gasoline AH, the diesel AH had overall smaller EFs, and the effect of operating temperature is not as straightforward as with gasoline. There was a slight increase of PN>3.4 nm by a factor of 1.2 in laboratory conditions, which is almost entirely due to increases in spikes contribution to overall emissions as stable operating $EF_{min}$ are almost zero with its ratio of laboratory to outdoor $EF_{min}$ being only 2.3 %. For diesel, the PN>11 nm decreased by factor of 0.44, whereas the PN>22 nm did not change due to operating temperature change. For both the PN>11 and the PN>22 there has similarly been a shift in the temporal composition of emission with more pronounced spikes and smaller stable operating emissions. The relative contribution of spikes to preheating cycle PN emissions were in the range of 53-60 %, and 98-99 % for outdoor measurements and laboratory measurements, respectively.

The effect of aging to PN emissions was assessed in the laboratory measurements with the environmental chamber. For gasoline AH there was no detectable new particle formation during the exhaust aging process, which resulted in a reduction of all measured PN $EF_{fuel}$s by 29 %, 31 %, and 4 % for 3.4, 11, and 22 nm cut-offs, respectively. The smaller decrease in





largest cut-off can be explained by coagulation removing mostly smaller particles and increasing GMD of particle size distribution and also condensation into the existing smaller particles growing them into size range detectable by largest cut-off CPC mitigating the effects of particle losses during the aging process. In contrast, for the diesel AH there was notable increase in PN EF$_{fuel}$s growing by factors of 10, 4.9, 5.1 for 3.4, 11, and 22 nm cut-offs, respectively due to the new particle formation in the relatively clean chamber (Fig. 8c–d). It should be noted that aging of PN emissions is extremely sensitive to experimental conditions. Thus, it is probable that if exhausts measured in these experiments would be released into the (same) clean outdoor air and allowed to dilute freely, that new particle formation would (more) similar between the diesel and gasoline AH exhausts.

### 3.2.2 particle size distributions

During the stable operation period of the gasoline AHs, the PSDs remained were relatively invariable (Fig. 4a−b), with average GMDs of measurements ranging from 21 to 24 nm. These results align well with results from previous outdoor measurements, where stable period GMD was found to be 21 nm for gasoline AHs and 16 nm for diesel AHs (Oikarinen et al., 2022). For the diesel AH, the number of particles detected during the steady period were low enough to cause larger variation in the geometric mean diameter during a preheating cycle (Fig. 4c−d). On average the detected particles were smaller than for the gasoline AHs, with average GMD ranging from 8 to 11 nm, for renewable and regular diesel, respectively. where 8 nm averaged GMD is from renewable diesel.

Considering changes in the PSDs during the preheating cycle, all AHs produced the largest particles during the ignition phase, while for the shutdown phase there is a clear downward shift in GMD. There are, however, notable differences in the degree of these shifts between vehicles and even repeated measurements with same vehicle, as can be seen from the gasoline AH measurements with a distinctly strong downward shift in the GMD during the first measurement (Fig. 4a). This stronger downwards shift in the first measurement also coincided with larger HC emissions, indicating enhanced new particle formation via nucleation in the tailpipe or during dilution.

Faster changes in PSDs during ignition and shutdown periods can be estimated by comparing total concentrations of CPCs with different cut-off diameters and relative fractions of size bins calculated from CPCs (Fig. S6). For both gasoline and diesel AHs PSD shows shift towards larger particles during ignition spike, when fraction of particles in size ranges of 11−22 nm and >22 nm grows rapidly. Shutdown spikes of diesel AHs were observed to behave similarly to ignition spikes with initial increase in >11 nm particles followed by stronger growth in concentration of sub-11 nm particles. Shutdown spikes for the gasoline AH meanwhile differed from ignitions spikes during the first seconds of shutdown spike, the fraction of >22 nm particles increases slightly, but with no corresponding increase in 11−22 nm fraction as was observed for ignition spikes. After those initial seconds, the shutdown spike of gasoline AH behaved similarly to other observed spikes with the increase in sub-11 nm fraction with coinciding reduction in larger fractions.





**Figure 4: Fresh PN size distribution measured by the SMPSs for the gasoline experiments (a-b), and renewable diesel (c) and regular diesel (d) experiments on the left side and PN concentrations measured by the CPCs with effective cut-off diameters of 3.4, 11, and 22 nm on the right side. Geometric mean diameters (GMD) of the SMPS size distributions are presented as black lines.**
**Due to overlapping size ranges of SMPS's, composite size distribution was formed by taking weighted average for concentration values in an overlapping portion of the size range.**

**Table 2. Mean concentrations, emission factors per kg of consumed fuel, for the whole 30-minute cycle, for 1 minute of stable operation, and the fraction of emissions emitted during the spike periods (ignition and shutdown) for the particle emissions, with standard deviations of respective values inside brackets. The chamber-aged EFs from**
**laboratory measurement campaign are also presented. Aged to fresh ratios were calculated from changes in**





concentrations in the chamber. Measurements for which measured values were below device detection limit are marked with BD, whereas – indicates data not being available.

| Quantity (device) | Unit | Gasoline AH | | | | | Diesel AH | | | | |
|---|---|---|---|---|---|---|---|---|---|---|---|
| | | Outdoor | Laboratory | | Laboratory/Outdoor | Laboratory | Outdoor | Laboratory | | Laboratory/Outdoor | Laboratory |
| | | Fresh | Fresh | Aged | Fresh | Aged/Fresh | Fresh | Fresh | Aged | Fresh | Aged/Fresh |
| PN3.4 (CPC) | $10^6\,cm^{-3}$ | 8.4 (5.4) | 56 (27) | 40 (17) | 6.7 (5.3) | 0.71 (0.46) | 0.88 (-) | 1.5 (0.17) | 15 (2.5) | 1.7 (0.2) | 10 (2.1) |
| | $10^{12}\,kg\ fuel^{-1}$ | 180 (110) | 910 (440) | 640 (280) | 5.1 (4) | 0.71 (0.47) | 26 (-) | 33 (3) | 330 (46) | 1.2 (0.11) | 10 (1.7) |
| | $10^{12}\,30\ min^{-1}$ | 46 (29) | 240 (120) | 170 (75) | 5.2 (4.1) | 0.71 (0.47) | 6.6 (-) | 8.2 (0.74) | 83 (12) | 1.2 (0.11) | 10 (1.7) |
| | $10^{12}\,min^{-1}$ | 1.4 (0.96) | 7.7 (3.7) | - | 5.5 (4.6) | - | 0.12 (-) | 0.0028 (0.00028) | - | 0.023 (0.0023) | - |
| | Spike % | 30 (1.1) | 12 (0.96) | - | 0.39 (0.035) | - | 53 (-) | 99 (0.26) | - | 1.9 (0.005) | - |
| PN6.3 (CPC) | $10^6\,cm^{-3}$ | 12 (-) | 18 (-) | 22 (-) | 1.5 (-) | 1.2 (-) | 0.74 (-) | - | - | - | - |
| | $10^{12}\,kg\ fuel^{-1}$ | 270 (-) | 290 (-) | 310 (-) | 1.1 (-) | 1.1 (-) | 27 (-) | - | - | - | - |
| | $10^{12}\,30\ min^{-1}$ | 72 (-) | 76 (-) | 83 (-) | 1.1 (-) | 1.1 (-) | 6.9 (-) | - | - | - | - |
| | $10^{12}\,min^{-1}$ | 2.1 (-) | 2.5 (-) | - | 1.2 (-) | - | 0.092 (-) | - | - | - | - |
| | Spike % | 27 (-) | 12 (-) | - | 0.45 (-) | - | 53 (-) | - | - | - | - |
| PN11 (CPC) | $10^6\,cm^{-3}$ | 4.8 (2.9) | 19 (2.1) | 15 (3) | 4 (2.5) | 0.77 (0.18) | 0.57 (0.14) | 0.36 (0.23) | 1.8 (0.11) | 0.64 (0.44) | 4.9 (3.2) |
| | $10^{12}\,kg\ fuel^{-1}$ | 110 (63) | 310 (32) | 210 (41) | 2.8 (1.7) | 0.69 (0.15) | 19 (4.1) | 8.1 (5.4) | 36 (1.2) | 0.44 (0.31) | 4.5 (3) |
| | $10^{12}\,30\ min^{-1}$ | 29 (17) | 82 (8.3) | 56 (11) | 2.9 (1.7) | 0.69 (0.15) | 4.5 (0.83) | 2 (1.4) | 9 (0.31) | 0.45 (0.31) | 4.5 (3) |
| | $10^{12}\,min^{-1}$ | 0.81 (0.53) | 2.7 (0.3) | - | 3.3 (2.2) | - | 0.054 (0.0028) | 0.00084 (0.00033) | - | 0.016 (0.0061) | - |
| | Spike % | 28 (2.8) | 11 (0.63) | - | 0.39 (0.046) | - | 53 (2.8) | 98 (1.9) | - | 1.8 (0.1) | - |
| PN22 (CPC) | $10^6\,cm^{-3}$ | 2.8 (1.9) | 12 (2.5) | 13 (2.9) | 4.3 (3) | 1.1 (0.33) | 0.27 (0.1) | 0.29 (0.18) | 1.6 (0.14) | 1.1 (0.79) | 5.5 (3.5) |
| | $10^{12}\,kg\ fuel^{-1}$ | 58 (37) | 190 (38) | 180 (40) | 3.3 (2.2) | 0.96 (0.28) | 6.9 (1.9) | 6.5 (4.3) | 33 (2) | 0.95 (0.67) | 5.1 (3.3) |
| | $10^{12}\,30\ min^{-1}$ | 15 (9.9) | 50 (10) | 48 (10) | 3.3 (2.3) | 0.96 (0.28) | 1.7 (0.39) | 1.6 (1.1) | 8.3 (0.49) | 0.99 (0.68) | 5.1 (3.3) |




| | | | | | | | | | | | |
|---|---|---|---|---|---|---|---|---|---|---|---|
| | $10^{12}$ min$^{-1}$ | 0.47 (0.33) | 1.7 (0.35) | - | 3.5 (2.6) | - | 0.02 (0.00098) | 0.00057 (0.00022) | - | 0.029 (0.011) | - |
| | Spike % | 27 (3.4) | 11 (0.69) | - | 0.4 (0.056) | - | 60 (5.4) | 99 (1.5) | - | 1.7 (0.15) | - |
| PM (ELPI+) | mg m$^{-3}$ | 0.48 (0.22) | 0.56 (0.11) | 3.9 (3.1) | 1.2 (0.59) | 7 (5.8) | 0.046 (0.015) | 0.052 (0.054) | 18 (25) | 1.1 (1.2) | 350 (610) |
| | mg kg fuel$^{-1}$ | 10 (4.5) | 9 (1.7) | 64 (51) | 0.89 (0.43) | 7.1 (5.9) | 1.4 (0.49) | 1.1 (1.2) | 400 (570) | 0.81 (0.89) | 360 (630) |
| | mg 30 min$^{-1}$ | 2.6 (1.2) | 2.4 (0.46) | 17 (13) | 0.89 (0.44) | 7.1 (5.9) | 0.33 (0.1) | 0.28 (0.29) | 100 (140) | 0.85 (0.92) | 360 (630) |
| | mg min$^{-1}$ | 0.075 (0.041) | 0.07 (0.015) | - | 0.93 (0.54) | - | BD | BD | - | - | - |
| | Spike % | 41 (4.6) | 20 (2.7) | - | 0.48 (0.084) | - | 88 (8.8) | 100 (0.0016) | - | 1.1 (0.11) | - |
| eBC (AE33) | mg m$^{-3}$ | 0.27 (0.12) | 0.8 (0.17) | 0.83 (0.16) | 3 (1.5) | 1 (0.3) | 0.013 (0.0056) | BD | 0.072 (0.095) | - | - |
| | mg kg fuel$^{-1}$ | 5.8 (2.5) | 13 (2.7) | 12 (2.2) | 2.2 (1.1) | 0.93 (0.26) | 0.37 (0.18) | 0.082 (0.011) | 1.5 (2) | 0.22 (0.11) | 18 (25) |
| | mg 30 min$^{-1}$ | 1.5 (0.65) | 3.4 (0.71) | 3.2 (0.57) | 2.3 (1.1) | 0.93 (0.26) | 0.087 (0.039) | 0.02 (0.0028) | 0.38 (0.5) | 0.24 (0.11) | 18 (25) |
| | mg min$^{-1}$ | 0.029 (0.014) | 0.1 (0.025) | - | 3.6 (1.9) | - | BD | BD | - | - | - |
| | Spike % | 46 (3.9) | 15 (2.7) | - | 0.33 (0.065) | - | 100 (0) | 100 (0) | - | 1 (0) | - |

### 3.2.3 Particle mass

Both gasoline and diesel AHs exhaust had clear peaks in the PM mass during the ignition and shutdown phases (Fig. 5). For
the gasoline AH, the freshly emitted PM mass was higher during shutdown phase compared to ignition phase, in contrast to
the diesel AHs for which the exceedance of the stable level was similar for both phases. During the stable operating period
fresh PM emissions of gasoline AH remained at elevated levels during both outdoor and laboratory measurements, especially
when compared against low stable period PM emissions of diesel AH.

During laboratory measurements for the stable operation period diesel AH did not produce enough PM mass to be detectable
with ELPI, which is in line with low PN concentrations. The PM mass was similar for both tested diesel fuel types during
steady phase. During ignition and shutdown phase, in contrast, the fossil-based diesel produced notably higher emissions
compared to the renewable diesel especially during ignition phase and this effect is more pronounced on the aged emissions.

The photochemical aging in TSAR increased the particle mass concentrations from the tailpipe by 2 to 3 orders of magnitude
in the laboratory conditions, and by factor of 3 to 30 in the outdoor measurements. The difference between fresh and aged



concentrations was especially prominent in the ignition and shutdown phases of the AH preheating cycle, as these were the phases with excessive emission of gaseous SOA precursors. SOA enhancement ratio was especially prominent in laboratory diesel measurements where primary PM mass is very low. $EF_{fuel}$s of fresh and aged PM mass were 9, and 1.1 mg $kg_{fuel}^{-1}$ for

fresh gasoline and diesel AH exhausts, respectively, and corresponding aged $EF_{fuel}$s were 64, and 400 mg $kg_{fuel}^{-1}$ for the aged exhausts. The results show how that low primary emissions do not necessarily result in low total PM when effects of aging are accounted for, and that the direct PM emission from the AH may pale in comparison to secondary PM emissions. The SOA formation potential and emission factors by photochemical aging in the TSAR were also compared to SOA formation in the environmental chamber; see Sect. 3.3.2.


When outdoor PM mass measurements are compared against laboratory measurements the fresh mass $EF_{30min}$s for the gasoline AH were found to be similar between outdoor and laboratory measurements (Fig. 9). The fresh $EF_{30min}$s from both gasoline laboratory tests fall within a single standard deviation of the $EF_{30min}$s determined from outdoor measurements. The fresh $EF_{30min}$s with the regular diesel were slightly over one standard deviation larger in the laboratory conditions compared

to the field conditions, whereas the fresh $EF_{30min}$ from renewable diesel were notably (factor 4.4) lower in the lab compared to the outdoor $EF_{30min}$s. Emissions for the renewable diesel were not measured in outdoor measurements, so direct comparison of operating temperature cannot be directly inferred. However, when compared against fresh $EF_{30min}$ of regular diesel during laboratory measurements, a clear decrease in fresh mass is observable.





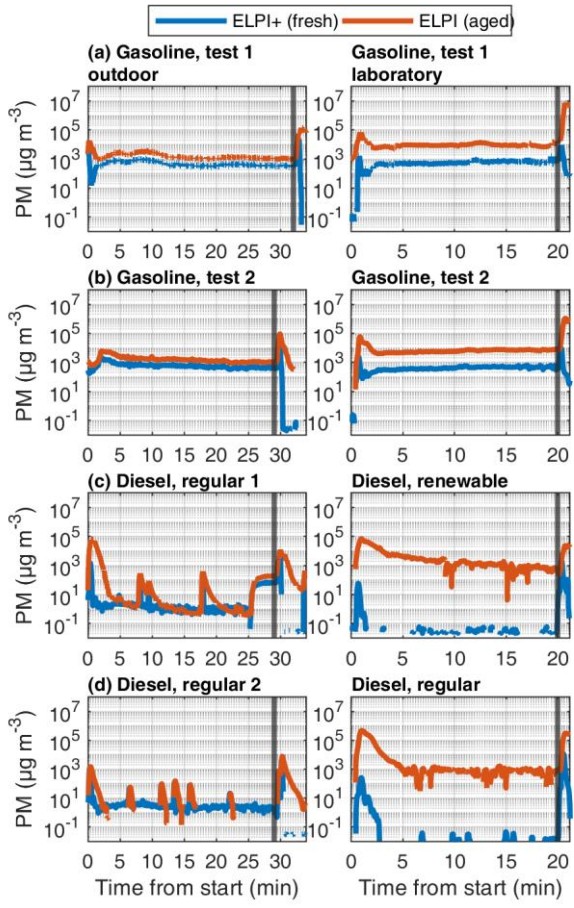

**Figure 5: Fresh and aged PM mass concentrations in raw exhaust emission, measured by ELPIs. The aged time series are adjusted by −20 s to account for the delay due to longer sampling line and TSAR.**

### 3.2.4 Black Carbon and Ångström exponent

The gasoline powered AH produced far greater eBC concentrations compared to the diesel-powered AH in both outdoor and laboratory conditions (Fig. 6). The eBC showed similar ignition and shutdown spikes and stable operating period shape as the total PM mass. For diesel AHs, the eBC concentrations were below the aethalometer detection limit, apart from the ignition and shutdown spikes in both outdoor and laboratory measurements. Changes in ambient temperature affected eBC emissions for both AH types. The EF$_{fuel}$s of the gasoline AH eBC were greater under laboratory conditions by a factor of 2.2 when compared to the outdoor measurements. For the diesel AHs, the effect of operating temperature was the opposite, and eBC EF$_{fuel}$s in laboratory conditions were reduced by 78 % when compared to the outdoor EF$_{fuel}$s.

The AAEs of the gasoline AH exhaust particles during stable operation were in the range of 1.06 to 1.53 during the outdoor measurements and in the range of 1.47 to 1.48 during the indoor measurements. For the diesel AH, AAEs could not be




reliably determined due to the low concentration of absorbing aerosols. For ignition and shutdown periods, the AAEs could

however be estimated for both vehicles when sufficient signals for eBC were detected (Fig. S8). During ignition, AAE

tended to increase roughly at the same time as eBC spiked. For gasoline AH, the highest AAE values were observed after the

end of the ignition spike in eBC which stabilized to values in the range of 1.7 to 2 together with the stabilization of the eBC.

The AAE of the diesel AH exhaust emission clearly decreased after ignition spike in the eBC spike. During shutdown AAE

decreased slightly at the beginning or just before shutdown spike formation, but no clear changes in the AAE were observed

afterwards. Gasoline AH AAEs returned to pre-decrease levels during shutdown period as long as clear eBC signal could be

observed, while the diesel AHs had decreasing trend in AAE during shutdown period with sharp decrease in AAE at the end

of measurement as eBC signal returned to near zero.

+ Impact of aging on AAEs?

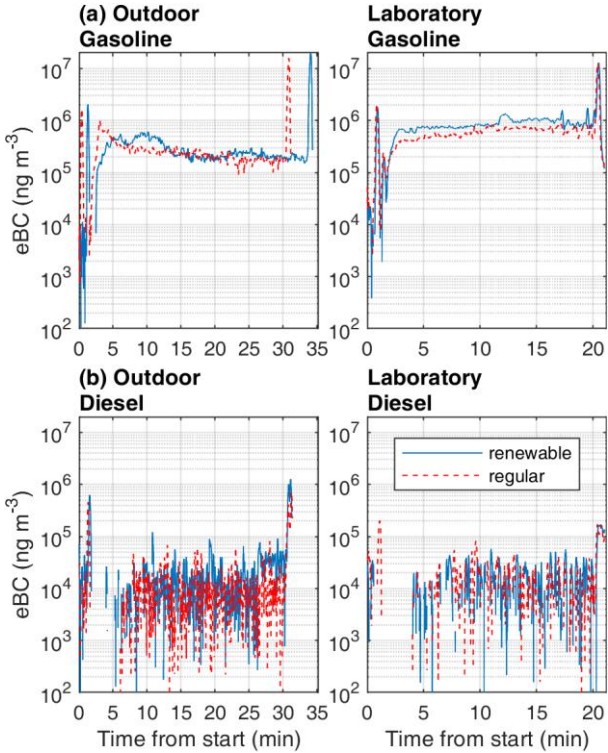


**Figure 6: Fresh black carbon mass concentrations of measured auxiliary heaters. On the left side are results from outdoor winter measurements and on the right side are results from indoor laboratory measurements. Laboratory diesel AH measurements were done using renewable and regular diesel fuels for all the other tests fuel was same for both tests. It should be noted that missing parts of time series indicate negative eBC-signal from measurement device.**

**3.2.5 Chemical composition**

The chemical composition of the fresh exhaust particles, measured by the SP-AMS, was dominated by rBC, followed by

organic matter (Fig. 7). The contribution by other constituents was minor. It should be noted that the composition presented





in Fig. 7 only includes particles with an aerodynamic diameter above ~50 nm due to the limitations of the SP-AMS, which represents 94 % to 96 % of the total particulate mass estimated based on ELPI. For the gasoline AH experiments, the rBC

measured by the SP-AMS was only 10 to 15 % of the eBC, whereas for the diesel experiments rBC was higher than eBC in the fresh exhaust (Fig. S9).

Organic matter became the dominating factor for the diesel AH after photochemical processing due to the high SOA formation relative to the primary soot emissions in the primary emission (Fig. 7). For the gasoline measurement, the fraction

of rBC in the secondary emissions was higher than for the diesel due to lower SOA enhancement ratio. In addition to organic matter, there was a slight increase in the fraction of sulfates (4%) for gasoline AHs, but not for diesel AH exhausts.

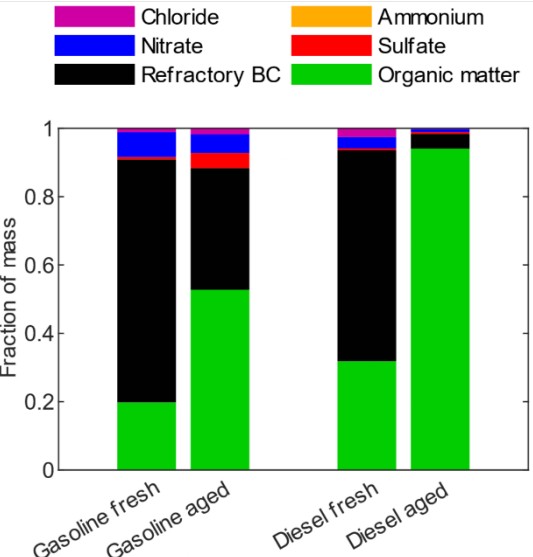

**Figure 7: Mean chemical compositions of the fresh and aged AH particle mass fractions for gasoline and diesel measurements.**


**3.3 Development of the photochemically aged exhaust in the chamber**

**3.3.1 Particle formation and growth**

The first test for gasoline AH (Fig. 8a) showed no new particle formation events and only fairly linear increase in GMD from 32 nm to 47 nm. Increase in [OH] by added $H_2O_2$ for the gasoline test 2 (Fig 8b) led to notably faster particle growth, with

GMD changing from 30 nm to 60 nm. Nevertheless, the particle size distribution remained unimodal and log-normally distributed (Fig. 8b). The impact of coagulation on the particle growth was minor compared to condensational growth, with an estimated 3 to 4 nm increase in diameter for the gasoline experiments by coagulation only. The impact of coagulation on



the particle population was however evident, as it explained most of the decrease in particle number in the chamber. For diesel AHs, coagulation was determined to be negligible due to the low initial particle concentration.


The diesel AH exhaust experiments exhibited a similar increase in original distributions particle sizes. Additionally, new particle formation occurred at roughly 40-minute mark (Figs. 8c–d). For fossil diesel, a second event happened around 200-minute mark after addition of $H_2O_2$, where growth of smaller particles was observed (Fig 8d). Both the final amount and growth rate of new particles was greater for fossil-based diesel's emissions.




**Figure 8: Development of the wall loss corrected particle number size distributions in the chamber as measured by the SMPSs (left), and momentary size distributions at selected times after the UV lights were turned on (right). t = 0 corresponds to moment the chamber UV-lights were turned on.**

### 3.3.2 Secondary particle mass formation potential

For the TSAR aged $EF_{30min}$s, gasoline test 1 (Fig. 9) showed slightly greater deviation from the outdoor mean, with deviation just slightly greater than 2 standard deviations. The aged PM mass $EF_{30min}$s of the diesel AH, in contrast, depended on the type of fuel used. The TSAR aged PM mass $EF_{30min}$s of the diesel exhaust were order of magnitude greater in laboratory conditions when compared to outdoor measurements. It should be noted that although the fresh PM mass $EF_{30min}$ was smaller for the regular diesel, the aged $EF_{30min}$s were greater for the renewable fuel and his applies to both TSAR and chamber aging. Further the laboratory measurement $EF_{30min}$ values provided in Fig 9. differ slightly from the ones provided in Table 2, since $EF_{30min}$s presented in fig. 9 were calculated from ELPI data whereas values in Table 2 are calculated with ELPI+, this was done so that PM values for both outdoor and laboratory measurement are calculated from the same ELPI device.

Chamber aged $EF_{30min}$s were 5 to 10 times smaller than TSAR aged $EF_{30min}$s with the renewable diesel measurement being a major outlier with 1.4 times greater chamber aged $EF_{30min}$ than TSAR aged $EF_{30min}$ (Fig. 9). This might be due to differing chemical composition of both particles and secondary aerosol precursors produced by combustion of renewable diesel having different reactivity to rapid ozone and UV-light induced aging process of TSAR, whereas in the chamber aging SOA formation pathways might differ.

In the measurements where $H_2O_2$ was injected into the chamber, more SOA formation occurred in chamber than in the TSAR. It is especially noteworthy that after $H_2O_2$ boosted chamber aging, the detected PM concentrations decreased if chamber aged sample is also aged with TSAR (Fig. 9). Without $H_2O_2$ addition chamber aging roughly doubled PM $EF_{30min}$s when compared to fresh emissions for gasoline and regular diesel. $H_2O_2$ addition caused further increase by factor of 4.2 and 12 for gasoline and diesel AH exhausts, respectively. For the renewable diesel increase was by factor of 630, and $H_2O_2$ caused further increase by factor of 3.6. It is assumed that the chamber aged EFs are most representative for SOA formation from exhaust emissions under normal atmospheric aging conditions, whereas TSAR provides information on the maximal SOA formation potential under extremely favorable SOA formation conditions.

Relative variances of the measured fresh and aged $EF_{30min}$s were compared to estimate similarities of differently aged EF measurements between fuel types and consistency of repeated measurements with the same fuel. The directly measured fresh $EF_{30min}$s had the smallest relative variance of 1.2 between repeated measurements and different fuel types. Of the aged $EF_{30min}$s, the TSAR aged $EF_{30min}$ has the smallest relative variance of 4.6. In contrast, the chamber aged $EF_{30min}$, $H_2O_2$




boosted chamber aged $EF_{30min}$, chamber and TSAR aged $EF_{30min}$ having relative variances of 35, 113, and 72, respectively.

610  Meaning that despite relatively small variances in fresh $EF_{30min}$s between measurements the aging process with chamber or TSAR increases the variability of PM emissions. It should be noted that renewable diesel measurement is a major outlier from the rest of the measurements. If renewable diesel was excluded from the calculations, then the relative variances would be significantly lower with 0.74, 3.8, 3.7, 5.4 and 0.2 for fresh $EF_{30min}$, TSAR aged $EF_{30min}$, chamber aged $EF_{30min}$, $H_2O_2$ boosted chamber $EF_{30min}$, and chamber and TSAR aged EF30min, respectively.

615

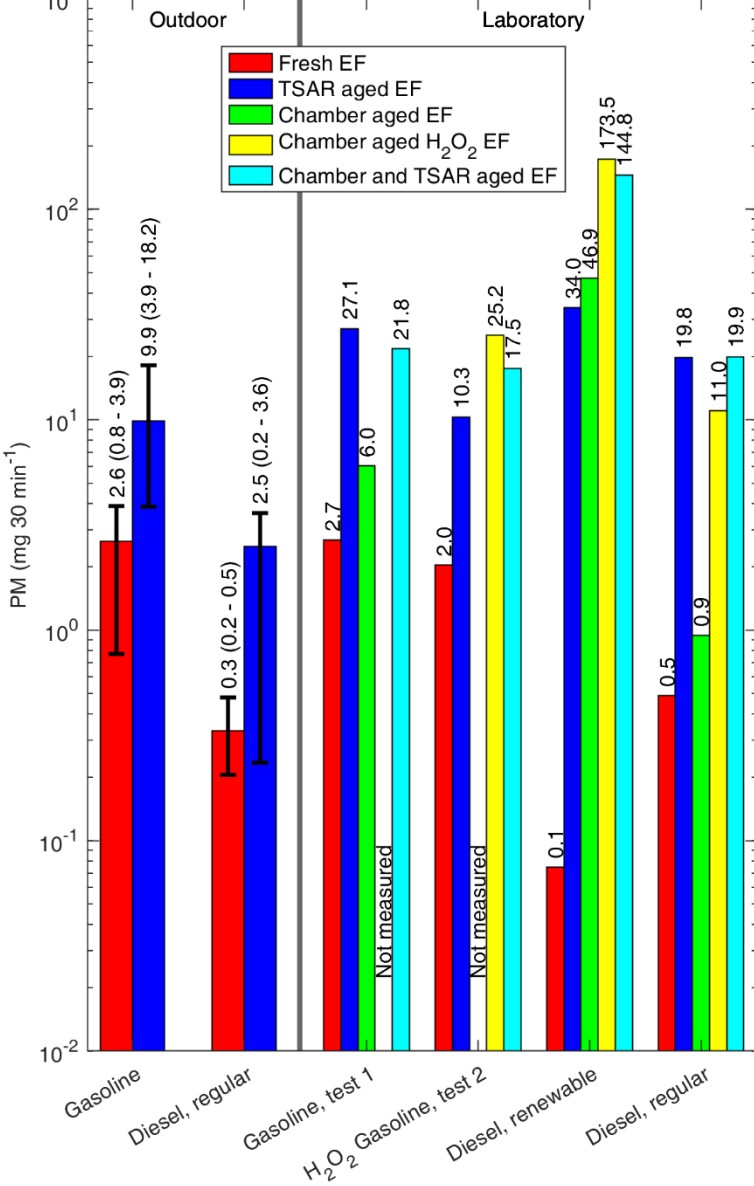



**Figure 9: Fresh and aged emission factors of particle masses for both outdoor and laboratory measurements. During Outdoor measurements Fresh EF$_{30min}$ was measured with ELPI+, and TSAR aged EF$_{30min}$ was measured simultaneously with ELPI. For laboratory measurements all particle mass concentrations were measured with ELPI. For outdoor measurements, a range of four repeated measurements are presented with error-bars.**

## 4 Conclusions

A multifaceted analysis of the emissions produced by auxiliary heaters was conducted in order to comprehensively address the environmental concerns arising from their use in passenger cars. This study utilized a combination of controlled outdoor experiments and laboratory investigations to provide a holistic understanding of the emission profiles AH systems operated with common fuels. By comprehensively analysing emissions under various conditions and fuel scenarios, this research contributes to a deeper understanding of AH emissions in atmospheric context.

We explored the impact of the operating temperature on emission profiles, mainly the role of laboratory and Finnish winter conditions (−19 to −7 °C), uncovering distinct effects across various emission components. For most emission components, the cold environmental temperature usually led to lowered emission factors of exhaust pollutants, however there are some contradictory observations. One important feature of the cold combustion conditions is the denser intake air which due to ideal gas law which naturally leads to higher air-to-fuel mass ratios. This can enable cleaner combustion conditions.

Our investigation revealed the importance of start and shutdown events on emission rates, and further disparities between gasoline and diesel emissions, during steady-state conditions where significant particle emissions persist in gasoline engines. This significant effect of ignition and shutdown to AH emissions indicates that AH heating is less polluting when it is done in longer continuous periods to avoid excessive on-off switching producing emission spikes. The effect of frequent ignitions is especially important for those vehicles where AH heating is periodically needed also during driving.

Of the regulated gaseous emissions, the CO concentrations exceeded the regulation limits of 1000 ppm for gasoline AH. This especially should be taken account when the AHs are used in closed environments, like unheated parking garages. THC and OGC concentrations stayed under regulation limits during stable operation but during ignition and shutdown the limits were exceeded.

The detailed quantification of gaseous and particle emissions included assessment of the role of secondary aerosol formation from the AH exhausts. Our findings highlight the emergence of aged aerosols, specifically secondary organic aerosol formation, under simulated photochemical OH-exposure conditions. These aged aerosols were seen to surpass fresh PM mass emissions by 1 to 3 orders of magnitude and effect of aging was found to be especially prominent during the ignition and shutdown phases of AH operation. This indicates that main atmospheric PM effects of AH are released in the gas phase from the exhaust line. This gives the indication that e.g. functioning oxidation catalysts could be a solution for secondary



organic PM reduction. Further research is still needed to quantify which specific chemical and physical processes are main drivers of secondary organic aerosol formation from AH exhaust.

We emphasize the need for continued research to enhance our understanding of emissions of fuel-operated auxiliary heaters at vehicle fleet level to more accurately quantify their effects on air quality and global road transport emissions. There is the need for revising the emission regulation for auxiliary heating devices. As with the current legislation, it is possible to surpass the emission limits defined for vehicles with the auxiliary heater taking care of additional heating for the engine and cabin, as AH emissions are not counted as part of regulated vehicle emissions. The current specific regulations of AHs are also less restrictive than vehicle emission regulations. Less restrictive regulation combined with AH emissions not being 660 counted as part of vehicle emissions allows for greater total emissions from vehicle use, if heat required by the vehicle is generated in a separate AH unit as opposed to generating the same heat with an internal combustion engine.

**Author contributions**

**HO:** Conceptualization, Methodology, Formal analysis, Investigation, Software, Visualization, Writing – original draft, Writing – Revised manuscript. **AH:** Formal analysis, Visualization, Writing – original draft preparation, Writing – review & 665 editing. **PS:** Investigation, Data curation, Formal analysis, Writing – original draft preparation, Writing – review & editing. **MO:** Conceptualization, Data curation, Investigation, Methodology, Project administration, Resources, Writing – review & editing. **U-VM:** Formal analysis, Visualization, Writing – review & editing. **PeM:** Investigation. **LS:** Investigation, Writing – review and editing. **VS:** Investigation. **SMa:** Conceptualization, Data curation, Investigation. **JH:** Investigation. **MI:** Investigation. **PaM:** Investigation. **PY:** Investigation. **OS:** Investigation, Writing – Revised manuscript. **SMi:** 670 Conceptualization, Methodology, Funding acquisition, Project administration, Supervision, Writing – original draft, Writing – Revised manuscript. **PK:** Conceptualization, Investigation, Methodology, Funding acquisition, Project administration, Supervision, Writing – original draft, Writing – Revised manuscript.

**Competing interests**

The authors declare that they have no conflict of interest.

**Financial support**

This research has been supported by the Jane ja Aatos Erkon Säätiö (project AHMA), the Tampere Institute for Advanced Study (Tampere IAS), the Kone Foundation, Henry Fordin Säätiö (grant no. 20230042), and the following Research Council of Finland (RCoF) grants: Competitive funding to strengthen university research profiles (PROFI) for the University of



Eastern Finland (grant no. 325022 and 352968), Flagship programme "ACCC" (grant nos. 337550, 337551, 359343,
357903), and Academy Research Fellow (grant no. 354226).

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
