# Peer review of "The role of fuel and environmental conditions on the amount and composition of primary, fresh, and aged aerosol emissions originating from diesel- and gasoline-operated auxiliary heaters of passenger cars"

_EGUsphere, 2025_

## Author Comment (AC1)

We thank both reviewers for the careful reading and valuable comments. Below we provide our point-by-point responses to the reviewers' comments. In the following context, raised comments/suggestions are marked in **black**, responses are presented in **green**, and changes to the manuscript/supplement information are indicated in **blue**.

In addition to suggestions by the reviewers, we have made some minor technical changes to the manuscript, e.g. removed reference to In Prep. Manuscript by Ala-Hakuni et al. and fixed some typos.

**Reply to Anonymous referee #1**

The authors measured the emissions of air pollutants from fuel-operated auxiliary heaters (AHs) in passenger cars. They estimated the emission factors (EFs) both outdoors (−19 to −7 °C) and in the laboratory (+25 °C), and evaluated the temperature dependence of the EFs. They also estimated secondary emissions using an environmental chamber and an oxidation flow reactor. In general, this manuscript is well written, and provides useful information on the emission characteristics of AHs, which are not well understood to date. However, I have several concerns as listed below.

We thank the referee for positive remarks about our manuscript.

(1) The authors evaluated the EFs of secondary pollutants. As well known (e.g., Hallquist et al., 2009; doi:10.5194/acp-9-5155-2009), formation yields of secondary organic aerosol (SOA) are strongly affected by the air temperature, exposure to oxidants (e.g., OH exposure), NOx levels, concentrations of seed particles, and so on. In this manuscript, the EFs of secondary pollutants are summarized in Figure 9, but the experimental conditions are not properly reported. I recommend the authors to state the air temperature, OH exposure, and NOx levels (or NOx/HC ratio) of individual experiments for fair comparison.

For example, the emissions of fresh PM mass and OGCs from the renewable diesel case were greater than those from the regular diesel case, whereas the aged PM mass was greater for the renewable fuel (L584-585). Without the information on the experimental conditions, we cannot assess whether this difference is explained by the different experimental conditions or by the different precursor composition.

Air temperature, photochemical age calculated based on OH exposure, and NOx concentrations in the ILMARI chamber for all 4 measurements are listed on the table below which has been added to section 2.1.2. OH-exposure information was not available for all measurement days due to malfunctioning of the VOCUS-PTR which was used to estimate the OH-exposure by decay of either butanol-d9 (for diesel AH) or toluene (for gasoline AH). Despite photochemical age not being available for all days we have added table of known conditions of the aging chamber to section 2.1.2. In addition, we expect that photochemical ages in gasoline, test 2 and diesel, regular are closer to 40 h measured for renewable diesel than 4 measured for gasoline, test 1 due to observable new particle formation and growth of existing particles as can be seen from Fig. 8. Also following sentences have been added to the third paragraph of section 2.1.2:

However, due to malfunctioning of the of the VOCUS-PTR used to estimate the OH-exposure the value for photochemical age is known for only single gasoline and diesel experiment of the two conducted for each fuel type.

Known aging conditions in the chamber are provided in the table 1 for photochemical age, OH exposure, NOx concentrations at both the start and end of aging experiment, and chamber temperature.

**Table 1: Aging chamber conditions for laboratory experiments. Photochemical age equivalent relative to 1.5 10^6 molec. cm-3 OH concentration, and corresponding cumulative OH exposure. Also, NOx concentrations at start and end of aging process and the temperature of the chamber are included.**

| Experiment | Photochemical age (h) | OH exposure ($10^{10}$ molec. cm$^{-3}$ s) | NOx concentration (ppb) at start and end of measurement | Air temperature (ºC) |
|---|---|---|---|---|
| Gasoline, test 1 | 4 | 2.1 | 53 - 44 | 21 ± 1.2 |
| Gasoline, test 2 | - | - | 56 - 11 | 21 ± 1.2 |
| Diesel, renewable | 40 | 26 | 36 - 12 | 21 ± 1.2 |
| Diesel, regular | - | - | 41 - 14 | 21 ± 1.2 |

(2) Standard deviations (SDs) of EFs for both outdoor and laboratory measurement are shown in Tables 1 and 2, but the SDs of EFs for laboratory measurement are not shown in Figure 9. I recommend the authors to clearly state the numbers of laboratory measurements under individual conditions and the meaning of SDs in Tables 1 and 2.

Range of measured EF values are not shown in the Figure 9 for laboratory measurements since each of measurement is represented separately as the experimental aging conditions were different for each measurement with either different fuel type used (Diesel) or different chamber ageing procedure with addition of $H_2O_2$ (gasoline), so range of EF values from multiple repeated measurements could not be provided as was done for the outdoor measurements where each of the 4 measurements for gasoline and diesel had similar fuel types and aging process. This has been clarified in the figure 9 text by amending figure's text:

Figure 9: Fresh and aged emission factors of particle masses for both outdoor and laboratory measurements. During Outdoor measurements Fresh $EF_{30min}$ was measured with ELPI+, and TSAR aged $EF_{30min}$ was measured simultaneously with ELPI. For laboratory measurements all particle mass concentrations were measured with ELPI. For outdoor measurements, a range of four repeated measurements are presented with error-bars. For laboratory measurements EFs of each individual experiment are presented separately.

Information about number of repeated measurements used to calculate SDs in tables 2 and 3 (note the changed table numbering due to addition of table 1) have been added to clarify this. Changes table texts:

**Table 2. Mean concentrations, emission factors per kg of consumed fuel, for the whole 30-minute cycle, for 1 minute of stable operation, and the fraction of emission emitted during the spike periods (ignition and shutdown) for the gaseous emission, with standard deviations of respective values inside brackets. Number of measured heating cycles used to calculate the means, and standard deviations were 4 for outdoor measurements and 2 for laboratory measurements for both gasoline and diesel AHs. Measurements for which measured values were below device detection limit are marked with BD, whereas – indicates data not being available.**

**Table 3. Mean concentrations, emission factors per kg of consumed fuel, for the whole 30-minute cycle, for 1 minute of stable operation, and the fraction of emissions emitted during the spike periods (ignition and shutdown) for the particle emissions, with standard deviations of respective values inside brackets. The chamber-aged EFs from laboratory measurement campaign are also presented. Aged to fresh ratios were calculated from changes in concentrations in the chamber. For the outdoor measurements number of repeated measurements used to calculate means and standard deviations for all instruments besides CPCs was 4, for the CPCs PN3.4 had 1 measurement for diesel AH and 2 for gasoline AH, while PN6.3 nm has 1 measurement for both fuel types. For the laboratory measurements number of repeated measurements used to calculate means and standard deviations were 2 for all instruments besides CPCs, for CPCs PN3.4 had 2 measurements for both AHs and PN 6.3 had 1 measurement for gasoline and 0 for diesel. Measurements for which measured values were below device detection limit are marked with BD, whereas – indicates data not being available and for measurements with only single measurement available, the lack of standard deviation is denoted with (-).**

(3) Information on aftertreatment devices on the AH (e.g., particulate filters, oxidation catalyst) is helpful to readers.

All measured AHs lacked aftertreatment devices. We clarify this further in the text by adding following information into section 2.1.1. Authors are not aware of any commercially available AHs with emission aftertreatment devices. Furthermore, it is likely that no AHs with aftertreatment devices exist due to relatively loose regulation of AH emissions not providing sufficient incentives to develop aftertreatment devices for AHs.

It should be noted that both measured AHs lacked emission aftertreatment devices such as particle filters and oxidation catalysts commonly used to reduce engine emissions of vehicles. However, lack of aftertreatment devices is typical for commercially available AHs.

Specific                                                                                                                         comments:

L251-253: Let me confirm the meaning of the "preheating". Is it the whole heating cycle of AH, including ignition and shutdown? I could not correctly understand the meaning of this word, and the clarification of this word is helpful.

Preheating refers to whole heating cycle including the ignition and shutdown. Only from the $EF_{min}$ are the ignition and shutdown parts of the preheating cycle excluded as stated later in the same paragraph on the line 262. Reference to preheating specifically was done due to common use case for AH being that you preheat your vehicle before driving

by turning AH on for about 30 minutes. It would be clearer to use heating cycle instead of preheating in this paragraph. Text has been modified to make this clearer as follows:

Emission factors (EF) of AHs were calculated for 1) over the whole 30 min heating cycle (EF$_{30min}$), 2) per 1 min of stable operation by excluding the ignition and shutdown parts of the heating cycle (EF$_{min}$), 3) EF per kg of fuel consumed during 30 min heating cycle (EF$_{fuel}$), and 4) fraction of total emission produced during ignition and shutdown spikes for both outdoor and laboratory measurement campaigns.

L350: Is this temperature dependence consistent with that of gasoline and diesel vehicles?

Temperature dependence of NOx emissions from gasoline and diesel vehicles has been observed in other studies. For example, in Olin et. al. 2023 reduction in the NOx emissions of between 30%-90% was observed for gasoline vehicles and between 1%-55% for diesel vehicles when vehicle was warm before driving. In that study emissions from driving 13.8 km route during winter conditions (-28...-10 °C) were measured for both cold started and warm initial engine temperatures, where largest reductions were observed when vehicle was warmed by driving 13.8 km once before measurements to get both engine and emission aftertreatment devices warm enough to operate efficiently.

Study by Wærsted et. al. 2022 also observed reduction in the NOx emissions when road traffic emissions were compared between warm and cold days from 46 roadside monitoring stations. They estimated that NOx emissions are on average 70 % lower at >14 °C when compared to –13 °C.

So, our observation that the NOx emissions of gasoline and diesel AH are reduced by about 33 % in warm laboratory temperatures compared to cold outdoor temperatures is consistent with results for vehicle emissions. Reason for relatively smaller reduction between warm and cold temperature NOx emissions from AHs is assumed to be related to lack of aftertreatment devices for NOx which might perform non-optimally in cold temperatures which increases the temperature dependency of NOx emissions from vehicles. Text in section 3.1.2 has been amended to include mentions about studies where temperature dependence of vehicle NOx has been measured.

The NO$_x$ EFs for both gasoline and diesel AHs were smaller by roughly one-third under laboratory conditions compared to outdoor conditions (Table 2). Observed temperature dependency of NOx emissions from AHs is consistent with studies of temperature dependency of vehicle NOx emissions. Where both warm initial vehicle temperature before driving (Olin et. al. 2023) and warm outdoor temperatures (Wærsted et. al. 2022) have been observed to correlate with lower NOx emissions when compared to cold temperatures.

L377: typo ("were had")

Typo was fixed by removing the word were from the sentence. Sentence now reads as:

For diesel AH, alkanes, aromatic hydrocarbons, and unsaturated hydrocarbons had similar concentrations during stable period and during the shutdown spike oxygenated HCs overtake unsaturated ones.

L458-460: Do you mean enhanced new particle formation from extremely low volatile organic compounds (ELVOCs) or from VOC? A simple explanation with references is helpful to the readers. (If you mean ELVOCs, is the comparison between the HC emissions and new particle formation significant?)

Text should state that the downwards shift coincided specifically with large organic gaseous carbon (OGC) emissions as can be seen from the blue line of the figure 3b. Referencing the term HC was indeed too ambiguous in this context without reference to previous figures. On closer inspection more likely explanation for differences in observed small particles during shutdown spikes of two gasoline tests would be that low time resolution of the SMPS causes incomplete detection of shutdown spike depending on which specific sections of PSD SMPS was scanning during shutdown.

Text has been changed to reflect this along with addition of references to figure 3b which referenced large coinciding OGC emission can be observed, alongside with explanation about problems caused by low time resolution:

This strong downwards shift in the first measurement also coincided with large OGC emissions (Fig. 3b, gasoline test 1), possibly indicating enhanced new particle formation via nucleation from oxidation products of extremely low volatile organic compounds in the AH exhaust either in the tailpipe or during dilution. However more likely explanation for differences between detected PSDs during shutdown spikes would be that low time resolution of SMPS of 3 minutes relative to duration of shutdown spike lasting 1-2 minutes makes it hard to accurately compare differences in shutdown spikes. As the differences between two gasoline tests in Fig. 4a-b could also be caused by timing differences of PSD scans resulting in different parts of the spikes PSD being measured. This is supported by PN concentrations detected with high time resolution CPCs ($\sim$1 s$^{-1}$) with both gasoline tests showing similar concentrations during shutdown spikes for cut-off diameter of 11 nm, but for >3.4 nm second gasoline test (Fig. 4b) shows much higher concentration than first test (Fig. 4a) indicating higher concentration of 3.4-11 nm particles should be observable for the second test than the first. As this is not the case the SMPSs have insufficient time resolution to accurately compare rapid changes in PSDs during ignition and shutdown spikes of AH heating cycles.

L499-501: "9 and 1.1 mg kg-1": Is this for fresh mass or for "fresh and aged mass"? I could not correctly understand the meaning of "the fresh and aged PM mass" for "fresh AH exhausts".

9 and 1.1 mg kg-1 are both for fresh masses. Original sentence was written ambiguously and has been clarified to make clear that the first 2 EFs are for fresh masses of gasoline and diesel respectively and later 2 EFs are aged EFs of gasoline and diesel. Sentence now reads as:

$EF_{fuel}$s of fresh PM mass were 9, and 1.1 mg $kg_{fuel}^{-1}$ for gasoline and diesel AH exhausts, respectively, and corresponding aged $EF_{fuel}$s were 64, and 400 mg $kg_{fuel}^{-1}$.

L538: typo ("+ Impact of aging on AAEs?")

Typo left over from older draft has been removed from the manuscript

---

## Author Comment (AC2)

We thank both reviewers for the careful reading and valuable comments. Below we provide our point-by-point responses to the reviewers' comments. In the following context, raised comments/suggestions are marked in **black**, responses are presented in **green**, and changes to the manuscript/supplement information are indicated in **blue**.

In addition to suggestions by the reviewers, we have made some minor technical changes to the manuscript, e.g. removed reference to In Prep. Manuscript by Ala-Hakuni et al. and fixed some typos.

**Reply to Anonymous referee #2**

The manuscript by Oikarinen et al. reports experimental data for emission of both gas and particulate matter species from fuel-operated auxiliary heaters. The experiments include operation of the heater under a winter-time condition, making the study to be relevant to understand the real environment. Overall, the experiments seem to be well-conducted, and the data are well organized. The dataset would be useful for understanding the urban environment. Though the manuscript is a good report of data, all the employed measurement methods are well-established ones (i.e., contribution to the advancement to atmospheric measurement technique is unclear). The reviewer is not sure if the manuscript fits well with the scope of the journal. The reviewer leaves this concern to the editor. Regarding the main content of the manuscript, I only have a few minor comments.

We thank the referee for positive remarks about our manuscript. Regarding the subject matter of this manuscript being outside the scope of AMT journal. We believe that the presented dataset, although based mainly on established methods, contributes to the atmospheric measurement community by providing novel and comprehensive real-world data on a rarely studied but relevant emission source. However, we fully respect the editorial decision regarding suitability for AMT.

Line317

The authors report emission of HCl. Is Cl included in fuel, or other material of the heater?

We expect that the Cl is present in the fuel rather than material of the heater. Despite that as chemical composition of the fuel was not analysed separately in this study, we cannot say that with certainty. Possible sources for trace amounts of chlorine present in the fuels might be impurities leftover from crude oil used to refine fuels or due to contamination of the fuels with chlorine-based cleaning agents used to clean fuel tanks of gas stations or tanks used to transport the fuel from refinery to gas station as reported by Battaglia et al., 2025. HCl was measured due to its corrosive potential as gaseous hydrogen chlorine can form strong hydrochloric acid when it interacts with water. Brief description of HCl emission has been added to section 3.1.2 to address the presence HCl emission in more detail.

HCl concentrations of 0.27 ppm and 0.26 ppm were measured for gasoline and diesel AHs respectively with corresponding $EF_{fuel}$s of 6.6 mg $kg_{fuel}^{-1}$ and 8.9 mg $kg_{fuel}^{-1}$. HCl was mostly released evenly during whole heating cycle with only spikes detected for gasoline AH during shutdown. Possible sources of HCl emissions from AHs could be trace amounts of chlorine in fuel leftover from crude oil used to refine fuels or due to contamination of the fuel with

chlorine-based cleaning agents used to clean fuel tanks of gas stations or tanks used to transport the fuel from refinery to gas station (Battaglia et al., 2025).

Line336

Emission of SO2 is reported. Could the authors clarify the sulfur contents of the fuels that were employed for the experiment? It could influence the emission factor of $SO_2$.

Specific sulfur contents of the measured fuels were not verified independently based on chemical analysis of the fuel, but European Union has regulation limiting the maximum sulfur content of the fuel to 10 mg/kg$_{fuel}$. Measured $SO_2$ EFs are assumed to correspond to mass fraction of sulfur content in the fuels used in AH experiments. Ratio of molar masses of $SO_2$ to sulfur (64.066 g/mol for SO2, and 32.06 for Sulfur, so ratio is ~2) can be used to convert $SO_2$ EF$_{fuel}$s of exhaust to estimates for sulfur content of the fuel assuming that detected $SO_2$ originates solely from combustion of fuel.

Converting EF$_{fuel}$s to estimates of fuel sulfuric content of gasoline 2.6 mg/kg$_{fuel}$ would be well below regulation limit of maximum allowed sulfur content of 10 mg/kg$_{fuel}$, but diesel would be above allowed limit with 29 mg/kg$_{fuel}$, with regular diesel specifically having 14 mg/kg$_{fuel}$ and renewable diesel having higher estimated content of 44 mg/kg$_{fuel}$. Even for more conservative estimate where EF$_{fuel}$ of $SO_2$ is calculated from only from stable operation period emissions the sulfur content of the diesel fuel would still be on average be 24 mg/kg, with regular diesel specifically having 10 mg/kg$_{fuel}$ and renewable diesel having higher estimated content of 37 mg/kg$_{fuel}$.

Following addition has been made to section 3.1.2 about estimates of fuel sulfur content based on $SO_2$ measurements.

$SO_2$ emissions from AHs are not directly regulated, but European Union has regulation limiting the maximum sulfur content of commercial fuels used in road vehicles to 10 mg kg$_{fuel}$$^{-1}$ (Directive, 1998). Assuming that all detected of $SO_2$ originates solely from combustion of sulfur present in the fuel EF$_{fuel}$ of $SO_2$ can be used to estimate sulfur content of the fuel. The fuel sulfur contents corresponding to gasoline and diesel AH measurements would be 2.6 mg kg$_{fuel}$$^{-1}$ and 29 mg kg$_{fuel}$$^{-1}$ respectively, which would be well below the regulation limit for gasoline but above the limit for diesel. Of the two types of diesel fuel measured the renewable diesel had higher $SO_2$ emissions with corresponding estimates for fuel sulfur contents being 44 mg kg$_{fuel}$$^{-1}$ and 14 mg kg$_{fuel}$$^{-1}$ for renewable and regular diesel, respectively. Even for more conservative estimate where EF$_{fuel}$ of $SO_2$ is calculated from only from stable operation period emissions the sulfur content of the diesel fuel would still be on average be 24 mg/kg, with regular diesel specifically having 10 mg kg$_{fuel}$$^{-1}$ and renewable diesel having higher estimated content of 37 mg kg$_{fuel}$$^{-1}$. Fuel sulfur contents higher than 10 mg kg$_{fuel}$$^{-1}$ allowed by regulation has also been observed for similar renewable diesel as was used in this study based on chemical characterization of the fuel, where fuel sulfur content of 11 mg kg$_{fuel}$$^{-1}$ reported (Karjalainen et. al., 2019). So similar renewable diesel as was used in this study having higher than regulation limit of sulfur is not unprecedented.

Line380

It is interesting that usage of the renewable diesel influences the emission of OGCs. Could the authors add further descriptions about the cause?

We thank the reviewer for the comment. The fuels used were commercial pump-quality fuels, and similar (diesel) fuel types, though not from the same batches, have been characterized in detail in previous work (Karjalainen et al., Environ. Sci. Technol. 2019).

The observed lower OGC emissions with renewable diesel are likely related to its chemical composition and combustion characteristics. This renewable diesel contains almost no aromatic compounds and consists mainly of paraffinic hydrocarbons, while conventional fossil diesel contains significant amounts of aromatics ($\sim$30% w/w). Aromatic compounds are known to be more resistant to complete combustion, leading to higher emissions of unburned or partially oxidized organic species.

In addition, the higher cetane number and heating value of renewable diesel promotes more efficient combustion, leading to reduced formation of organic gaseous compounds. The improved combustion characteristics of paraffinic renewable diesel fuels have also been observed before. https://pubs.acs.org/doi/full/10.1021/acs.est.9b04073

Following additions have been made to section 3.1.3 regarding possible effects of diesel fuel type to OGC emissions:

The observed lower OGC emissions with renewable diesel are likely related to its chemical composition and combustion characteristics. While chemical composition of used fuels was not independently verified in this study it has been characterized in detail for similar diesel fuel types as were used in these experiments (Karjalainen et al., 2019). Based on that renewable diesel fuel used in these experiments contains almost no aromatic compounds and consists mainly of paraffinic hydrocarbons, while regular fossil diesel contains significant amounts of aromatics (28.6 wt %). Aromatic compounds are known to be more resistant to complete combustion, leading to higher emissions of unburned or partially oxidized organic species. In addition, higher cetane number and heating value of renewable diesel compared to regular diesel enables more efficient combustion, leading to reduced formation of organic gaseous compounds due to incomplete combustion of hydrocarbons present in the fuel.

Line 369

The reported particle number concentration is in the range of 10^7 # cm-3. Could the coincidence issue influence particle counting by the CPC for this high concentration range? If so, how much?

While the reported particle number concentrations were in the range of $10^7$ # cm$^{-3}$, which would be over the upper detection limit of the single count mode of CPCs of $5\times10^5$ cm$^{-3}$. This was however not issue with our measurement setup as sample was diluted before measurement with CPCs. When dilution ratio is accounted for even the highest single PN concentration detected during the measurements by CPCs was $1.1\times10^5$ cm$^{-3}$, which is well within the upper detection limit of our CPCs. Section 2.1.1 has been amended to mention effect of bridge diluter to CPCs specifically.

Additional dilution step was also applied for battery of condensation particle counters with bifurcated flow diluter with DR of 158 to avoid exceeding the upper detection limit of the single count mode of the condensation particle counters.

Line 430

Units are missing.

Clarification has been added to what EFs specifically are referenced when the operating temperature dependent changes in PN emissions for diesel AH are discussed. Text has been amended to read as follows:

There was a slight increase in $EF_{fuel}$ of PN>3.4 nm by a factor of 1.2 in laboratory conditions, which is almost entirely due to increases in spikes contribution to overall emissions as stable operating $EF_{min}$ are almost zero with its ratio of laboratory to outdoor $EF_{min}$ being only 2.3 %. For diesel, the $EF_{fuel}$ of PN>11 nm decreased by factor of 0.44, whereas the $EF_{fuel}$ of PN>22 nm did not change due to operating temperature change.

Figure 7

It would be interesting to show organic mass spectra, and discuss the data in detail.

Following paragraph on organic mass spectra has been added to the section 3.2.5. Figure of the spectra has also been added to the supplement.

A more detailed composition of the organic fraction can be seen in Figure S11. The fresh exhaust for both gasoline and diesel experiments are dominated by a strong signal at m/z 29 and a typical repeating pattern for hydrocarbon like organic aerosol (HOA) with signal at m/z 41, 43, 55 and 57. These results are similar to previous studies of vehicle exhaust (Mohr et al., 2009). The secondary aerosol mass spectra are similar to that of oxygenated organic aerosol (OOA) with an increased signal at m/z 44 and a clear shift to more oxygenated species. The secondary emissions are also what could be expected from secondary aerosol of vehicle exhaust (Mohr et al., 2009; Zhu et al., 2021). These results indicate that AH emissions can be difficult to distinguish from engine exhaust emissions, and for example finding a factor relating to AH emissions with positive matrix factorization could be difficult without clear differences from engine exhaust mass spectra.

[Figure]

Supplementary Figure 11. Normalized organic mass spectra of (a) gasoline fresh emissions, (b) diesel fresh emissions, (c) gasoline aged emissions and (d) diesel aged emissions.